# Emerging Trends in Immunotherapy for Cancer

**DOI:** 10.3390/diseases10030060

**Published:** 2022-09-06

**Authors:** Alok K. Mishra, Amjad Ali, Shubham Dutta, Shahid Banday, Sunil K. Malonia

**Affiliations:** 1Department of Molecular, Cell and Cancer Biology, UMass Chan Medical School, Worcester, MA 01605, USA; 2MassBiologics, UMass Chan Medical School, Boston, MA 02126, USA

**Keywords:** FDA, checkpoint inhibitors, monoclonal antibody, bispecific antibody antibody drug-conjugate, CAR-T, CAR NK, Trastuzumab, Enhertu, PD-1, PDL-1, BiTEs, oncolytic virus, cytokine therapy, cancer vaccine

## Abstract

Recent advances in cancer immunology have enabled the discovery of promising immunotherapies for various malignancies that have shifted the cancer treatment paradigm. The innovative research and clinical advancements of immunotherapy approaches have prolonged the survival of patients with relapsed or refractory metastatic cancers. Since the U.S. FDA approved the first immune checkpoint inhibitor in 2011, the field of cancer immunotherapy has grown exponentially. Multiple therapeutic approaches or agents to manipulate different aspects of the immune system are currently in development. These include cancer vaccines, adoptive cell therapies (such as CAR-T or NK cell therapy), monoclonal antibodies, cytokine therapies, oncolytic viruses, and inhibitors targeting immune checkpoints that have demonstrated promising clinical efficacy. Multiple immunotherapeutic approaches have been approved for specific cancer treatments, while others are currently in preclinical and clinical trial stages. Given the success of immunotherapy, there has been a tremendous thrust to improve the clinical efficacy of various agents and strategies implemented so far. Here, we present a comprehensive overview of the development and clinical implementation of various immunotherapy approaches currently being used to treat cancer. We also highlight the latest developments, emerging trends, limitations, and future promises of cancer immunotherapy.

## 1. Introduction

The concept of harnessing the immune system’s capability to eradicate cancer was conceptualized in the late 19th century by William B. Coley, also known as the “Father of Immunotherapy”. Coley’s experiments involving injections of live bacteria *S. pyogenes* and *S. marcescens* (also referred to as Coley’s toxin) into patients with inoperable cancers showed favorable responses [1,2,3,4]. Following this, Paul Ehrlich’s hypothesis in 1909 that the human body constantly generates transformed cells that are eliminated by the immune system, as well as Thomas and Macfarlane’s conceptualization of the ‘cancer immune surveillance’ hypothesis, laid the groundwork for future understanding of cancer immunology [4]. Understanding immune surveillance and editing processes have resulted in a new paradigm in cancer treatment. The immune system plays a dual role in protecting against tumor growth by activating innate and adaptive immune mechanisms while also shaping tumor immunogenicity [5]. Over the last two decades, there has been substantial progress in our understanding of how tumors manipulate the immune system, culminating in the development of innovative treatments that prevent tumor immune evasion. Allison and Honjo’s discovery of T-cell immune checkpoints CTLA-4 and PD-1, which awarded them the Nobel Prize in Physiology or Medicine in 2018, propelled the cancer immunology field into the present era of cancer immunotherapy. The last decade has witnessed the development of many immunotherapeutic approaches to cancer. Preclinical and clinical research continues to uncover new ways to harness the immune system’s ability to cure cancer and broaden the indications for currently available treatments. This review elaborates on various immunotherapies currently being applied for cancer treatment (Figure 1A). These immunotherapeutic approaches are (1) immune checkpoint inhibitors, an approach designed to ‘unleash’ T cell responses; (2) adoptive cellular therapies, which are based on delivering engineered immune cells into the body to fight cancer; (3) oncolytic virus, which selectively kills cancer cells without damaging normal cells; (4) cancer vaccines, which train the immune system to fight cancer; (5) cytokine therapies, an immunomodulatory approach; and (6) Monoclonal antibodies. We also discuss preclinical and clinical advancements of various immunotherapies, limitations, and emerging trends.

## 2. Cancer Immunity and Immune Evasion

The immune system has intricate monitoring mechanisms that detect neoplastically transformed cells in the body and elicit an appropriate immune response. This process is called immune surveillance [6,7]. However, emerging tumors employ multiple ways to evade immune-mediated clearance through several evolving molecular mechanisms. Immune interactions that contribute to tumor surveillance and growth are collectively referred to as immunoediting [6,7]. Immune surveillance and immunoediting rely on three phases: elimination, equilibrium, and escape [8] (Figure 1A).

During the elimination phase, the innate and adaptive immune systems recognize and respond to tumor-specific antigens and destroy tumor cells [8]. By continuously removing newly transformed cells, this phase basically demonstrates the fundamental idea of cancer immune surveillance [7]. However, some sporadically evolving tumor cells can survive in the elimination phase and enter a dynamic state of equilibrium [9]. The equilibrium phase is defined as a latency period between the elimination phase’s completion and the onset of clinically recognized malignant disease [6,10]. Next, during the escape phase, some of the tumor cell variants that emerged from the equilibrium phase acquire the capability to proliferate in an environment with a functional immune system and establish an immune suppressive environment that further aggravates the disease condition [5,6]. Tumor immune escape can occur either through intrinsic or extrinsic mechanisms. The intrinsic mechanisms include the changes in tumors themselves, weak expression of antigens in the initial stages of tumor growth, loss of epitopes, the physical barrier preventing effector cells from penetrating solid tumors, loss of antigen presentation, and production of immune inhibitory signals such as immune checkpoints [9,10,11]. On the other hand, tumor extrinsic mechanisms are conferred by the host immune system, including immune tolerance, anergy of tumor-specific T cells, production of soluble ligands that limit lymphocyte activation, and impairments of professional APC antigen presentation and maturation [6,7,9,10,11]. The concept of immune escape is currently being harnessed in multiple therapeutic settings, including targeting critical immune checkpoints in difficult-to-treat cancers.

## 3. Immune Checkpoints

The immune system is continually challenged by exogenous immunogens or endogenous immunogens produced by altered or transformed cells. The immune system’s effector components work continuously to eradicate the causative agents of both endogenous and exogenous immunogens and maintain immunological homeostasis [5,12,13]. A feedback control system termed immune tolerance prevents collateral damage caused by the hyperactivated immune response [14,15,16]. Immune checkpoints are a set of molecular effectors of the immune tolerance system that maintain immune homeostasis and avoid autoimmunity [10,17]. The checkpoint proteins are expressed on the surface of immune cells and help them distinguish between self and non-self by interacting with their cognate ligands [5,13,14,18]. Tumor cells may take advantage of this homeostatic mechanism through the immunoediting process by expressing ligands of immune checkpoints [12,15,18]. These immune checkpoints play a critical role(s) in cancer immunity and have been exploited as major therapeutic targets for cancer immunotherapy (Figure 2A). Some of the well-studied adaptive and innate immune checkpoints are discussed below:

### 3.1. Adaptive Immune Checkpoints

#### 3.1.1. CTLA-4

Cytotoxic T lymphocyte antigen 4 (CTLA-4) is CD28 related cell surface protein expressed on activated T cells [19,20]. It competes with CD28 for binding to ligands CD80 and CD86 present on antigen-presenting cells (APCs). When CTLA-4 binds to CD80/86, it generates a T-cell inhibitory signal by reducing the availability of CD-28 ligands. When CTLA-4 is overexpressed on CD8^+^ and CD4^+^ T cells it blocks CD28-B7 binding and inhibits stimulatory signals produced by MHC and TCR interaction [21]. Moreover, CTLA-4 is constitutively expressed on regulatory T cells (Treg) and is critical in regulating immunological self-tolerance [22]. The first anti-CTLA-4 monoclonal antibody (mAb), Ipilimumab (Yervoy) was approval by the U.S. FDA in 2011 for the treatment of unresectable or metastatic melanoma [23]. Since its approval, Ipilimumab has been used in combination with other checkpoint inhibitors, such as Nivolumab (anti-PD-1), to treat various cancers, including metastatic unresectable melanomas with or without BRAF mutation [24,25,26], microsatellite instability-high (MSI-H) or mismatch repair deficient (dMMR) colorectal cancer (CRC) [27,28], hepatocellular carcinoma (HCC) [29], malignant pleural mesothelioma [30], non-small cell lung cancer (NSCLC) [31] and renal cell carcinoma (RCC) [32].

#### 3.1.2. PD-1

Programmed death protein 1 (PD-1), also called CD279, is an inhibitory receptor expressed on activated T cells. It regulates T cell effector functions in various physiological responses, including chronic infection, cancer, and autoimmunity [33]. The interaction of PD-1 with its ligands PD-L1 (Programmed death ligand 1), also called CD274 and PD-L2 (CD273), mediates PD-1 signaling. [33]. The ligands of PD-1 are typically expressed on antigen-presenting cells such as macrophages, dendritic cells, B-cells, myeloid cells, and cancer cells [21,34,35]. Studies have shown that PD-1-PDL-1 interaction results in T-cell-mediated immune suppression under normal physiological conditions; however, cancer cells harness this mechanism to evade immune-mediated clearance [12,33]. When PD-1 binds to APCs or cancer cells, it prevents pro-inflammatory processes such as T cell proliferation and cytokine response leading to an immunosuppressive tumor microenvironment [13,15].

To date, six monoclonal antibodies (mAbs) targeting the PD-1/PDL-1 axis have been approved by U.S. FDA for various solid tumors and hematological malignancies [14]. Approved PD-1 mAbs include Pembrolizumab, Nivolumab, Cemiplimab, and Durvalumab, while PDL-1 mAbs include Atezolizumab and Avelumab [14,15]. In addition, the combination therapy comprising-anti-CTLA-4 and anti-PD-1 has shown promising results in various preclinical studies and is currently being tested in clinical trials [14,15,36]. The U.S. FDA first approved Nivolumab (anti-PD-1) and Ipilimumab (anti-CTLA-4) in 2015 for melanomas, and it was recently approved for esophageal cancer in 2022. Figure 2B depicts various FDA-approved regimens targeting PD-1/PDL-1 axis in various cancers.

#### 3.1.3. LAG-3

Lymphocyte activation Gene-3 (LAG-3) is a CD4-related surface receptor present on activated CD4^+^, CD8^+^ T cells, and regulatory T cells [37,38]. LAG-3 signaling inhibits T-cell activation, expansion, and cytokine production, leading to exhaustion [39]. It binds to the MHC-II complex and inhibits the interaction of MHC-II with CD4, resulting in reduced TCR signaling and an attenuated immunological response [39]. However, whether MHC-II alone is responsible for the inhibitory activity of LAG-3 is still unclear [40,41]. FLG1, FGL2 (Fibrinogen-like protein 1, 2), LSECtin (Lymph node sinusoidal endothelial cell C-type lectin), Gal3 (Galectin-3), and FREP1 are the other known LAG-3 ligands. The binding of LAG3 to these ligands leads to inhibition of T cell activation [40,41,42]. Studies have shown that LAG-3 is co-expressed on T cells along with other inhibitory molecules such as PD-1, TIGIT, and TIM-3; it promotes greater T-cell exhaustion than LAG-3 alone [43]. In cancer, LAG-3 expressing Tregs accumulate at distinct tumor sites, thereby suppressing cytotoxic T cells. Moreover, LAG-3 expression correlates with poor prognosis in many cancers [44,45]. Blockage of LAG-3 signaling could be used to activate anti-tumor immunity; however, inhibition of the LAG-3 pathway alone has been ineffective [46,47]. Preclinical studies have shown that blockade of the PD-1 pathway upregulates LAG-3 and combinatorial inhibition of LAG-3 and PD-1-PDL-1 axis in cancer exhibit durable antitumor response [48,49]. Following these preclinical developments, the U.S. FDA approved a combination immunotherapy regimen, Opdivo (Opdualag). This combination includes Relatlimab (anti-LAG-3 mAb) and Nivolumab (anti-PD-1 mAb), targeting both the LAG-3 and PD-1 for the treatment of unresectable advanced melanoma [50,51,52,53]. Opdualag is also being evaluated in clinical trials for many other cancers, such as liver, lung, and colorectal cancer [46,47,54]. LAG-3 mAb as monotherapy is also being investigated in esophageal or gastric cancer, multiple myeloma, and chordoma [46,47,55].

#### 3.1.4. TIGIT

T cell immunoreceptor with Ig and ITIM domain (TIGIT) is an Ig superfamily receptor that plays an important role in suppressing innate and adaptive immune responses. [56,57]. It is expressed in several immune cell types, including regulatory T cells, cytotoxic T cells, natural killer (NK) cells, and follicular T helper cells [56,57]. TIGIT indirectly lowers T-cell activation, which is crucial for restricting innate and adaptive immunity [56]. The three nectin and NECL molecular family proteins CD155(PVR), CD112(PVRL2 or Nectin-2), and CD113 (Nectin-3), which are expressed on tumor cells or antigen-presenting cells, are the known TIGIT ligands [58,59]. Interaction of TIGIT with its ligands suppresses immune activation [60,61]. The inhibitory effect of TIGIT is compensated by the immune-activating receptor CD226 (DNAM1), also expressed on cytotoxic T-cells and NK cells, and it competes with TIGIT binding to CD155 and CD112 [61,62]. The elevated presence of TIGIT and its cognate ligands results in T cell exhaustion and suppresses DNAM1 signaling culminating in the loss of T cell function [62,63].

Tumor cells exploit the inhibitory TIGIT pathway to escape immune-mediated destruction [56]; therefore, targeting TIGIT could be another strategy for cancer immunotherapy. Multiple TIGIT inhibitors are already in clinical trial stages [52,56,57]. BMS-986207, an anti-TIGIT mAb, is in phase I/II clinical trial for advanced solid tumor as monotherapy and in combination with Nivolumab or Ipilimumab (NCT04570839, NCT02913313). Another phase I/II randomized trial tested the efficacy of anti-TIGIT and anti-LAG3 mAbs in patients with relapsed refractory multiple myeloma either alone or in combination with pomalidomide and dexamethasone (NCT04150965, NCT02913313). Ten other TIGIT-targeting mAbs, including a phase III drug, Vibostolimab, are in clinical trials. In addition, another TIGIT-targeting antibody, Tiragolumab, in combination with Tecentriq (anti-PDL-1 mAb), has received a breakthrough therapy designation by the US FDA for the treatment of non-small cell lung carcinoma (NSCLC) that express high PDL-1 [64]. Given the number of TIGIT-targeted inhibitors that are now undergoing clinical studies, it may soon become another validated immune checkpoint.

#### 3.1.5. TIM-3

T-cell immunoglobulin and mucin domain-containing protein-3 (TIM-3), also known as HAVCR2, is an immune checkpoint receptor present on a variety of immune cells, including dendritic cells (DCs), natural killer (NK) cells, cytotoxic T cells and regulatory T cells (Tregs) and [65]. TIM-3 is a type-I membrane protein that acts as a negative regulatory immune checkpoint by suppressing both adaptive and innate immune responses [65,66]. Increased TIM-3 expression on T and NK cells is linked to exhaustion [65]. TIM-3 bind to numerous ligands, some of which include CEA cell adhesion molecule 1 (CEACAM1), Galectin-9, phosphatidylserine (PtdSer), and high mobility group protein B1 (HMGB1) [67,68,69,70]. Galectin-9, however, is considered the natural ligand of TIM-3 [67]. The co-expression of TIM-3 and CEACAM1 is associated with T cell exhaustion [65,66]

Evidence for TIM-3 as an immune checkpoint in cancer came from preclinical cancer models showing the CD8^+^ T cells co-express TIM-3 and PD-1. Dual expression of TIM-3 and PD-1 on CD8^+^ T cells cause more defects in effector function and cytokine production than PD-1 expression alone [71,72]. These findings suggested that TIM-3 may cooperate with PD-1 pathways, resulting in cancer’s dysfunctional phenotype of CD8^+^ T cells. [71]. Moreover, in human patients with advanced melanomas, non-small cell lung cancer (NSCLC), and Hodgkin lymphoma, approximately a significant percentage of CD8^+^ tumor-infiltrating lymphocytes express both TIM-3 and PD-I [71]. Clinical studies have shown that blocking TIM-3 and PD-1 has a concomitant synergistic effect on reducing tumor burden [73,74].

#### 3.1.6. B7-H3 and B7-H4

The B7 is a transmembrane protein that binds to the CD28 or CTLA-4 and modulates T cell-mediated immune signaling. B7-H3 (CD276) is expressed on various immune cells, including T cells, B cells, dendritic cells (DCs), and natural killer (NK) cells [75,76]. Whereas B7-H4 is ubiquitously expressed in immune cells, however, its robust expression is primarily rescrticted to activated T cells B-cells and monocytes and dendritic cells [77]. Both B7-H3 and B7-H4 are over-expressed in several solid and hematologic malignancies, and their expression is associated with poor prognosis in many cancers, including renal cell carcinoma (RCC), colorectal cancer (CRC), and non-small cell lung cancer (NSCLC) [75,76,77].

Clinical trials using monoclonal antibodies against B7 molecules have been conducted. A phase I/II clinical trial was done using an antibody-drug conjugate (ADC), MGC018, where Duocarmycin was conjugated with the monoclonal antibody targeting B7-H3. However, MGC018 was found to be toxic with various adverse effects [78]. Another phase-I therapeutic trial using naked monoclonal anti- B7-H4 antibody (FPA150) in B7-H4 expressing solid tumors yielded acceptable outcomes with moderate toxicities [78].

#### 3.1.7. VISTA

V-domain Ig suppressor of T cell activation (VISTA) is a type I transmembrane glycoprotein, also known as B7-H5, Dies1, PD-1H, Gi24, encoded by VSIR gene in humans [79]. It is mainly expressed on dendritic cells, macrophages, and myeloid cells, while its expression is relatively low in CD4^+^, CD8^+^ T cells, and Treg cells [79,80]. VISTA has characteristics with the B7 and CD28 family proteins and may function as a ligand and a receptor [81,82]. The known ligands of VISTA are P-selectin glycoprotein ligand 1 (PSGL-1) and V-set and Ig domain-containing 3 (VSIG3). The binding of VISTA to PSGL-1 is known to occur at an acidic pH, such as in the tumor microenvironment (TME); however, at physiological pH, VISTA-expressing cells can bind to PSGL-1. Interactions of VISTA with its ligands lead to attenuation of T cell function. [83,84]. VISTA is a key negative immune checkpoint regulator, which locks T cells in a quiescent state. It has been shown to inhibit T cell responses *in vitro* and in preclinical models of autoimmunity and cancer [85,86,87]. The role of VISTA as an immune checkpoint is demonstrated in studies using VISTA knockout (*Vsir*^−/−^) mice that exhibit exacerbated T cell responses and develop spontaneous autoimmunity [88]. The preclinical studies using several mouse models have demonstrated that VISTA inhibition leads to an increased T cell infiltration, proliferation, and effector activity in the tumor microenvironment [87]. The immunosuppressive role of VISTA on both lymphocytes and myeloid cells, as well as the its widespread expression on TILs, indicate that VISTA-blockade approaches may have broad therapeutic significance [80,83]. Three mAbs, VSTB112, P1-068767 (BMS-767), and SG7 targeting VISTA are in developmental phases. These antagonistic mAbs inhibit VISTA interaction with human PSGL-1 and VSIG3 with comparable potency [89]. Several preclinical studies have evaluated the effectiveness of VISTA targeting antibodies; however, most models have relied on a combination approach. VISTA and PDL-1 combination therapy in mouse models of colon cancer (CT26) and melanoma (B16) showed significant tumor regression and long-term survival [85]. Ongoing clinical trials for various anti-VISTA antibodies, such as a JNJ-61610588 and HMBD-002, are currently under investigation. HMBD-002 is in the clinical trial for advanced solid malignancies alone or in combination with Pembrolizumab, an anti-PD1 monoclonal antibody (NCT05082610). The phase I clinical trial has evaluated the safety and pharmacokinetics of JNJ-61610588 in patients with advanced cancers (NCT02671955). An oral small molecular inhibitor, CA-170 targeting both VISTA and PD1, has shown well-tolerance in phase I clinical trials for advanced solid tumors and lymphomas (NCT02812875) [80,81,90]. VISTA has shown the potential to become another target for immune checkpoint therapy. However, more research is warrented to delineate the role of VISTA across all cancer types. Several phase I and II trials of anti-VISTA therapy are underway, and further results from these trials will define the immunotherapeutic potential of VISTA in cancer.

#### 3.1.8. OX40/OX40L

The OX40 receptor (CD134) is a membrane-bound glycoprotein that belongs to the tumor necrosis factor (TNF) receptor superfamily. Its interaction with the OX40 ligand (a type II transmembrane protein) expressed on B cells, DCs, and endothelial cells, play a critical role in activating the effector function of CD4+ T-helper cells [91,92]. OX40 and CD28 signals have strong synergistic effects on the survival and proliferation of CD4+ T cells [93]. OX40L interaction with its receptor OX40 increases the expansion and survival of naive CD4+ T cells and memory Th cells by inhibiting peripheral deletion [94,95,96]. OX40-OX40L interaction plays a crucial role in T cell activation and improves T cell-mediated anti-tumor immunity, resulting in tumor regression and increased survival [97]. OX40 agonist antibodies have shown promising results in preclinical cancer models, including lung, colon, and breast cancer, as well as murine breast (4T1) and melanoma (B16) cancer models [97,98]. There are several ongoing clinical trials of OXO40 agonists, notably MEDI0562, a humanized OX40 agonist mAb is in phase I clinical trials for advanced solid tumors (NCT02318394). Anti-OX40 mAb, MEDI6469 as neoadjuvant therapy, is being investigated for advanced solid tumors in phase I as monotherapy and for B- cell lymphomas in phase II, in combination with other immunotherapeutic monoclonal antibodies (Tremelimumab, Durvalumab, and Rituximab) (NCT02205333). Neoadjuvant therapy is defined as the administration of therapeutic agents before the start of the main treatment [95]. MEDI6469, an anti-OX40 mAb, has been evaluated for safety and feasibility as a neoadjuvant treatment before surgery in patients with head and neck squamous cell carcinoma (HNSCC) [99]. Moreover, an engineered human OX40 Ligand IgG4P Fc Fusion Protein (MEDI6383) is also in phase I for recurrent metastatic solid tumors (NCT02221960). Although therapeutic targeting of OX40 has demonstrated promising results in preclinical settings, preliminary clinical data indicate its efficacy as monotherapy is limited. Combining OX40 co-stimulation with other immune checkpoint inhibitors like anti-PD-1 and anti-PD-L1 appears to be a promising strategy. However, further studies are required to better understand the co-stimulatory mechanism of OX40-targeted drugs in combinatorial therapy with other immune checkpoint inhibitors.

#### 3.1.9. A2A/B-R and CD73

In immune cells, CD73 (cluster of differentiation 73) dephosphorylates and transforms extracellular AMP to adenosine, which mediates its effects through the binding with the A2A or A2B adenosine receptor (A2AR or A2BR), which are expressed on T cells, APCs, NK cells, and neutrophils [100]. In the tumor microenvironment, high levels of ATP produced due to tissue destruction, hypoxia, and inflammation are catabolized by CD73, which is overexpressed in multiple immune cells present in the tumor environment as well as in multiple cancers [101,102]. Inhibitors of A2AR and CD273 have also reached clinical trials. A phase I trial was conducted using an oral inhibitor, EOS100850, targeting A2AR on T cells, as monotherapy for refractory solid malignancies [103]. Moreover, a monoclonal antibody targeting CD73 (CPI-006) has been evaluated as monotherapy or with an anti-A2AR drug (CPI-444) in the phase I trial [104]. Since targeting A2AR and CD273 is still in the early trial phases, more clinical trial data is needed to establish their therapeutic potential in cancers.

#### 3.1.10. NKG2A

Natural killer group protein 2A (NKG2A) is expressed on circulating NK cells and certain T cells and is enhanced in chronic inflammatory conditions [105]. NKG2A dimerizes with CD94 after interacting with its ligand HLA-E, inhibiting NK and T cell activity [105]. Elevated NKG2 expression has been linked to poor survival in ovarian and colon malignancies [93,94,95,96]. In the presence of other immune checkpoint inhibitors, such as anti-PD-1 or anti-PDL-1, blocking NKG2A was shown to enhance the anti-tumor response by T and NK cells [106,107,108]. Monalizumab, a humanized monoclonal antibody that targets NKG2A, has been evaluated as monotherapy in phase II clinical trials for recurrent or metastatic head and neck squamous cell carcinoma (HNSCC) [106,107,108]. In another phase II clinical trial on platinum reactant recurrent metastatic HNSCC patients, Monalizumab is also used in combination with Cetuximab, an EGFR-targeting monoclonal antibody [106,107,108]. More clinical studies are warranted to establish the overall efficacy of NKG2 inhibition in cancer.

### 3.2. Innate Immune Checkpoints

Immune checkpoints, as discussed above, target the adaptive immune system to induce a T cell-mediated antitumor response. However, the adaptive immune system works in tandem with the innate immune system to promote anti-tumor immunity [10,13,109]. Innate immune checkpoints, also known as “*Don’t eat me*” signals, are the molecular brakes that impede the interaction between tumor cells and the cells of innate immunity by restricting antigen presentation and antigen-specific anti-tumor response [13,109]. ‘*Don’t eat me*’ signals block the phagocytosis of cancer cells by macrophages and dendritic cells. Some examples of innate immune checkpoints have recently attracted attention as potential immunotherapy targets.

#### 3.2.1. SIRPα-CD47

IgSF family member SIRP*α* is a glycosylated transmembrane mostly expressed on macrophages, dendritic cells, and granulocytes [110,111]. However, except for T cells, it is expressed in several additional cell types of hematopoietic origin [109,111]. The primary SIRP*α* ligand is CD47 [109,110,112,113]. Upon binding to CD47, SIRP*α* is phosphorylated at its intracellular C-terminal ITIM motifs and triggers anti-inflammatory signaling in phagocytes [109,112,114]. This phosphorylation of SIRPα activates cytoplasmic phosphatases SHP-1 and SHP-2, which subsequently dephosphorylate Paxillin and non-muscle Myosin IIA, resulting in reduced phagocytosis [109,110,112,113]. Inhibition of SIRPα-CD47 interaction by anti-CD47 or anti-SIRPα antibody leads to a significant increase in cancer cell phagocytosis by macrophages and a decrease in tumor burden [109,112,113]. Even though normal human cells (including hematopoietic cells) frequently express CD47, blockade of CD47 or SIRPα preferentially causes phagocytosis of tumors because normal human cells lack prophagocytic or “eat me” signals [13]. Numerous approaches have been developed to target the CD47-SIRPα axis, including monoclonal antibodies targeting CD47 and SIRPα, recombinant SIRPα-fusion IgG, bispecific antibodies (BsAb), RNAi, small molecules inhibitors of enzymes that modify CD47 post-translationally and CD47-CAR T cell [13,109,114]. Several anti-CD47 humanized IgG4 mAbs, including Magrolimab (Hu5F9-G4), SRF231, Lemzoparlimab, and IBI188, are in different phases of clinical trials [13,114,115]. An engineered fusion protein, ALX142, which contains two high-affinity CD47-binding domains of SIRPα coupled to an inactive Fc region of human Ig, is also being evaluated in clinical trials for both solid and hematologic cancers [13]. However, the major limitation of anti-CD47 antibody-based therapy is dose-dependent phagocytosis of RBCs, leading to severe anemia in patients [13,113]. A priming dosage of the SIRPα-CD47 blocker or choosing the Fc domain of the anti-CD47 mAb to limit interactions with the phagocyte Fc receptor are currently being tried as strategies to reduce RBC phagocytosis and the associated anemia [13,109,112,113]. A recent review by Qu, T. et al. elaborates on numerous approaches for targeting the CD47-SIRPα axis and ongoing clinical studies of CD47 or SIRPα inhibitors [114]. The majority of CD47 inhibitors are still in phase I trials, except for a monoclonal antibody Magrolimab (in combination with Venetoclax and Azacitidine for T53 mutant AML) and a small molecule inhibitor RRx-001 (which targets both CD47 and SIRPα by downregulating their expression on cancer cells and macrophages respectively) for small cell lung cancer [114].

#### 3.2.2. LILRB1/MHC-I and LILRB2/MHC-I

Leukocyte Ig-like receptor B1 (LILRB1) is expressed in both innate and adaptive immunity cells, including macrophages, granulocytes such as eosinophils and basophils, dendritic cells, a subset of NK cells, as well as certain T and B cells [116]. Whereas MHC-I is expressed on nucleated cells and presents the processed antigen to the CD8^+^ T cells. Interaction of MHC-I to the T cell receptor is followed by a cascade of events leading to the activation of the cytotoxic function of CD8^+^T cells [47,116,117]. MHC-I is a heterodimer molecule made up of a heavy -chain and a β2-microglobulin (β2M) chain [117]. The expression of MHCI on cancer cells has recently been correlated with the resistance to phagocytosis through the interaction of its β2-microglobulin chain with LILRB1 receptors expressed on phagocytes [13,116,117]. However, it is worth noting that LILRB1 only interacts with MHCI via the β2M chain, as opposed to MHC-TCR, interacts which requires an intact MHC molecule [13]. Therefore, the approaches that precisely disrupt the LILRB1-β2M interaction can be potent innate immune targeting strategies [117]. In addition to this, another receptor of this family, LILRB2, which is also expressed on both innate and adaptive immune cells, also interacts with MHC-I. However, the specific ligand for LILRB2 is not yet known [13,117]. Inhibition of LILRB2 has been shown to promote the maturation and activation of macrophages [13,117]. A humanized anti-LILRB2 IgG4 mAb (JTX 8064), either alone or in combination with pembrolizumab (PD-1 inhibitor), has entered clinical trials (phase I, NCT04669899) for the treatment of advanced refractory solid tumors [13].

#### 3.2.3. Siglec10-CD24

Sialic acid binding Ig like lectin 10 (Siglec10), is predominantly expressed on innate immune cells such as macrophages [118,119], dendritic cells [120], and NK cells [121]. It acts as an innate immune checkpoint by interacting with its ligand CD24 present on tumor cells, leading to the functional inactivation of immune cells [118,122]. Multiple tumors have been shown to have altered Siglec10 expression [118], and an elevated expression in patients with kidney renal clear cell carcinoma (KIRC) negatively correlates with the survival of patients [118]. A recent study by Barkal et al. has demonstrated a crucial role of the Siglec10-CD24 axis in the evasion of cellular phagocytosis by tumor-associated macrophages [122]. Therapeutic inhibition of CD24 using monoclonal antibody leads to macrophage-dependent inhibition of tumor growth in vivo [122].

#### 3.2.4. APMAP

Adipocyte plasma membrane-associated protein (APMAP) is another novel antiphagocytic protein that has recently been discovered in a genome-wide CRISPR knockout screen for the identification of regulators of antibody-dependent cellular phagocytosis (ADCP) [123]. The study showed that the genetic depletion of APMAP in lymphoma cells significantly accelerates antibody-dependent cellular phagocytosis by human macrophages [123]. Furthermore, a counter CRISPR-knockout screen in J774 murine macrophage cells identified GPR84 (probable G-protein receptor) as a possible APMAP receptor on the macrophages [123]. However, follow-up studies are required to establish the mechanism by which APMAP may regulate antibody-dependent phagocytoses of cancer cells. Future studies focusing on the APMAP-GPR84 axis will shed more light on APMAP’s potential as an independent innate immune checkpoint in cancer. 

### 3.3. Trends in Clinical Trials with Immune Checkpoint Inhibitors

Clinical trials evaluating PD-1/PDL-1-blocking immunotherapies in combination with other immunotherapies, targeted therapy, chemotherapies, and radiation are steadily increasing, while clinical trials testing PD-1/PDL-1-blocking as a monotherapy are steadily declining. According to a recent analysis, 4062 out of 4897 ongoing clinical trials (approximately 83%) are evaluating PD-1/PDL-1 combination regimens in conjunction with various immunotherapies, targeted therapies, chemotherapies, and radiotherapies [35]. Nearly 300 targets and pathways are presently undergoing clinical trials in combination with PD-1/PDL-1 blocking immunotherapies to investigate new therapeutic avenues. Approximately 93 bispecific antibodies that target the PD-1/PDL-1 axis are currently under development. The majority of these are still in the preclinical stages. Bispecific antibodies are becoming increasingly popular in combination with PD-1/PDL-1 immunotherapy [35]. According to ClinicalTrials.gov, some of the selected phase II/II clinical trials using immune checkpoint inhibitors in combination with other therapies are shown in Table 1.

### 3.4. Limitations and Challenges of ICI Therapy

Immune checkpoint inhibitors (ICIs) have been employed as immunotherapeutic agents in treating various malignancies [124]. Immunotherapy has the potential to elicit long-lasting responses in a subset of patients with advanced diseases that can be maintained for several years after treatment cessation [125]. Compared to chemotherapy or targeted therapy, immune checkpoint inhibitor (ICI) therapy exhibits various tumor response patterns, such as delayed response, durable response, dissociated pseudoprogression, and hyperprogression [126]. One of the limitations of ICI therapies is the lack of reliable predictive biomarkers of durable response and limited understanding of clinically relevant determinants of pseudo or hyper progression. No standard definition of durable responses to ICI-based therapies has been devised, and optimal treatment duration in case of durable response has not been established [125,126]. However, a recent meta-analysis of randomized phase III trials assessing response to ICIs therapies indicated that the percentage of patients with a durable response was substantially higher with ICIs than with chemotherapy or targeted treatment [127]. Pseudoprogression is a rare phenomenon manifested in a subset of patients treated with ICIs, where an atypical pattern of tumor response is observed either after increased tumor burden or the appearance of new lesions [128,129]. Initially reported in a phase II trial evaluating the efficacy of Ipilimumab in metastatic melanoma [126,130], pseudoprogression has been observed in many clinical trials using ICIs for various solid tumors [128,129]. Hyperprogression, on the other hand, raises the possibility that ICI therapies may have the opposite effect on some patients. Hyperprogression is reported in 4–29% of patients, indicating that early clinical assessment is required for an effective treatment regimen [126]. The dissociated response has been observed small proportion of patients in which some lesions regress while others grow [126].

ICIs have shown remarkable success in inducing durable responses in several patients with malignant disease; however, these therapies confer distinct toxicities, depending on the type of therapy used [131]. Although acute toxicities are more frequent, chronic immune-related adverse events (irAEs), which happen in some patients, are becoming more recognized [131]. Immune-related adverse events (irAEs) are a distinct range of adverse effects of ICI therapies that resemble autoimmune reactions [132]. Clinical trial data analysis indicates that irAEs are estimated to occur in a significant proportion of cancer patients undergoing ICI therapies [23]. Since irAEs often result from immune system hyperactivation, this suggests that the exhausted immune cells have been reinvigorated to attack both tumor and the normal cells [131,132]. Studies have reported a positive and negative association between the occurrence of irAEs and the clinical response in cancer patients undergoing ICI-based therapies, but the results remain controversial so far [133]. The toxicities associated with checkpoint inhibitors, including pathophysiology, diagnosis, and management, have been extensively reviewed elsewhere [132]. Current research is focused on developing novel methods to mitigate toxicities and gaining a comprehensive understanding of various irAEs associated with checkpoint therapy [124]. Pharmacogenomic profiling of ICI-treated patients may provide greater insight into the critical genes and pathways mediating underlying toxicities. An important unresolved issue is the identification of biomarkers to predict response and treatment-mediated toxicity [124]. Therefore, finding biomarkers that can predict organ-specific toxicities associated with immune checkpoint therapies will be clinically beneficial.

Another limitation of ICI-based therapies is primary and acquired resistance. Although immune checkpoint therapy has been shown to have persistent response rates, many patients do not benefit from it, often called primary resistance. However, some responders experience a relapse of the metastatic disease after their initial response, also known as acquired resistance. Such types of heterogeneous responses have been observed in various metastatic lesions, even within the same patient [134]. This resistance is influenced by both extrinsic tumor microenvironmental variables and tumor intrinsic factors. The immunosuppressive tumor microenvironment created due to the presence of Tregs, M2 macrophages, MDSCs, and other inhibitory immune checkpoints largely contribute to acquired resistance [135]. Tumor resistance is also influenced by the absence of tumor antigen, loss or downregulation of MHC-I, alterations in the antigen-presentation, and inadequate immune cell infiltration. [134,136]. A recently proposed fitness model for tumors based on immune interaction with neoantigens can predict response to immunotherapy and can be useful in identifying acquired resistance to ICIs [137]. Recent advancements in immunotherapy research have resulted in developing novel delivery technologies and modifying existing antibodies to improve the side effects and therapeutic efficacy of immune checkpoint blockade therapies [124].

## 4. Adoptive Cell Therapy

Adoptive cell therapy (ACT) is another fast-emerging field of cancer immunotherapy in which a patient’s cells are genetically engineered *ex-vivo* and then transferred back to the patient’s body as therapeutic agents. T cell based adoptive cell therapies such as TILs (tumor-infiltrating lymphocytes), Synthetic TCRs (engineered T-cell receptors) and CAR T (chimeric antigen receptor T cells) and NK cells based therapies called CAR-NK have been developed. ACTs have made remarkable progress; as of 2022, almost 2756 active cell therapies are in various stages of development. There are more than 2500 estimated active cell therapy agents in the global oncology market, and CAR-T therapeutics continue to lead the cell-based therapy pipeline [138].

### 4.1. TILs (Tumor-Infiltrating Lymphocytes)

TIL therapy is an adoptive cell therapy that utilizes the patient’s naturally occurring T-lymphocytes infiltrating the tumor microenvironment [139,140]. The fundamental idea behind tumor-infiltrating lymphocyte (TIL) therapy is that while cytotoxic T cells that infiltrate the tumor microenvironment are exposed to tumor antigens and can attack tumor cells, they cannot completely eradicate the tumor because of their insufficient numbers [139,140,141]. In TIL therapy, tumor-infiltrating lymphocytes are isolated from the resected tumors, expanded ex vivo under activating conditions, and then reinfused into the patients to achieve a therapeutic anti-tumor response [139,140,142,143] (Figure 3A). 

Although no TIL therapy has received FDA approval so far, several TILs based therapies are in different stages of development [140,144,145]. Between 2011 and 2020, there were 79 trials of TIL therapy, comprising 22 different types of TIL products [144]. The autologous TIL products Lifileucels (LN144, LN145, LN145-S1) are currently in phase II of clinical development for recurrent or metastatic head and neck squamous cell carcinoma (HNSCC), relapsed or refractory metastatic non-small cell lung cancer (NSCLC) and unresectable or metastatic melanoma [140]. Moreover, the US FDA has recently granted breakthrough therapy designation to LN-145 for the treatment of advanced cervical cancer [140]. A breakthrough therapy designation is a process designed to expedite the review and development of a drug if the preliminary clinical evidence indicates that the drug may significantly outperform currently known treatment options for a serious diseases (www.fda.gov, accessed on 1 July 2022). A phase I clinical trial of TILs in anti-PD-1 resistant metastatic NSCLC patients reported that TILs in combination with lymphodepletion and IL-2 elicited a complete response with manageable toxicity [146]. A phase II clinical trial of TILs therapy report found a significant clinical response in patients with HPV-associated epithelial cancers, including metastatic squamous cell carcinomas and adenocarcinomas of the cervix [145]. This study provided evidence that TILs therapy is effective in treating epithelial cancers, which were previously thought to be a limitation of cell-based therapy. In addition, TILs in combination with anti-PD-1 have shown promising anti-tumor responses in multiple cancer types, notably PDL1 negative metastatic cervical cancer [141], metastatic osteosarcoma [147], triple-negative breast cancer [148].

TILs may offer a promising alternative to CAR T cells in solid and epithelial cancers, where CART cells have demonstrated limited efficacy despite remarkable success in B-cell malignancies [144,145]. Despite their effectiveness, TIL therapies have certain limitations, such as low in vivo persistence and restricted migration of the infused lymphocytes to the tumor site. Cancer immune evasion creates barriers to achieving the best clinical response [149]. Additionally, the administration of IL-2, which is given as a standard of care to sustain the survival and activity of TILs within the body, frequently results in systemic toxicity and necessitates close monitoring and care of patients undergoing TILs therapy. However, as lymphocytes in immunocompetent people are a primary source of cytokines that contribute to IL-2-associated side effects, IL-2 toxicities can be managed by adopting a lymphodepletion regimen and omitting high doses of IL-2 [150]. The development of next-generation TILs is mostly focused on genetically modifying TILs to overexpress cytokines like IL-2 and IL-12, which can provide a sustained activation and proliferation of infused lymphocytes [151]. The genetic depletion of negative regulators such as PD-1 [152] and CISH [153] is also being actively evaluated in preclinical studies to enhance the activity and efficacy of TILs. Additionally, the CRISPR-based genome-wide screening strategies such as CRISPR knockout, CRISPRi (interference), or CRISPRa (activation) are being pursued to identify potential targets whose deletion in modified TILs may yield the greatest benefits for TIL therapy [154,155,156,157,158].

### 4.2. TCR (T Cell Receptor) Therapy

TCR-based adoptive cellular therapy uses genetically manipulated lymphocytes that are targeted against specific tumor antigens. This approach utilizes the ability of TCRs to recognize the tumor-specific antigens presented by the major histocompatibility complex (MHC) present on the surface of malignant cells [159,160,161]. Typically, TCR-based therapy requires methodical and well-coordinated steps that include patient’s HLA (human leukocyte antigen) typing, selection of tumor-specific antigen, leukapheresis, manufacturing of transduced TCR product, lymphodepletion, and delivery to the patient [159,160,161] (Figure 3A). Most cell-based immunotherapies face challenges in delivering an effective pool of anti-tumor effector cells. However, the ex vivo production of up to billions of activated lymphocytes with known antigen selectivity and potency allows TCR therapy to overcome this obstacle [159,161]. TCR-based therapy utilizes a variety of immune cells, but T cells and NK cells are used most frequently [159]. Selecting a target-specific antigen (TSA) is the fundamental step in developing TCR-T cells. A specific antigen must be overexpressed on cancer cells compared to normal cells to prevent off-target and detrimental effects on healthy tissues. NY-ESO-1 (New York esophageal squamous cell carcinoma-1), which belongs to the cancer/testis (CT) antigens category, has been the target of most TCR-based therapy clinical trials to date, accounting for more than 35% of TCR product based clinical trials. CT antigens are a category of tumor antigens whose normal expression is restricted to male germ cells in the testis but not in adult somatic tissues [162]. In cancers, expression of CT antigen is elevated in a significant percentage of tumors of various types [162,163]. Other TSAs considered in TCR therapy include mutant antigens and neoantigens, most of which are safe targets due to their exclusive expression in tumor cells. TCR-engineered autologous T-cell therapy has demonstrated significant preclinical response in multiple myeloma [164], melanoma [165], and other solid tumors [166,167,168]. The U.S. FDA recently approved the first T cell receptor (TCR) therapeutic (tebentafusp) for patients with HLA-A*02:01-positive Uveal melanoma [169]. The approval of the therapy was based on a phase III trial including 378 patients receiving either tebentafusp, or the PD-1 inhibitor, pembrolizumab, the CTLA4 inhibitor, Ipilimumab, or the chemotherapy drug dacarbazine, or placebo. The study reported that recipients of tebentafusp had a 73% overall one-year survival rate, compared to 59% for the control group with some side effects [169,170]. Along with appreciable success, TCR therapy, like other emerging cell therapies, also has limitations and challenges associated with its widespread application. Genetically modified T cells are difficult to administer and are associated with significant long-term safety risks [159,171]. Moreover, due to the personalized nature of TCR therapy, several other inherent technical challenges are also found to be associated with the quality and acquisition of lymphocytes via leukapheresis and the manufacturing of TCR products [159]. TCR-based bi-specifics are also being developed as the next generation of TCR-based T cell engages due to the capacity of TCRs to bind a larger spectrum of antigens than bi-specific antibody-based T cell engagers [169].

### 4.3. CAR T Cells

CAR (Chimeric antigen receptor) T cells are modified primary human T cells engineered to express a chimeric antigen receptor (CAR) capable of recognizing tumor-specific antigens expressed on the surface of tumor cells [119,120,121]. CAR T-cell therapy has shown unprecedented growth in recent years and has become the most promising immunotherapy for B-cell-related malignancies [172,173,174]. The development of a chimeric antigen receptor CAR T cell therapy starts with collecting a patient’s blood and separating lymphocytes using a technique called apheresis or leukapheresis [175]. The apheresis product is then processed to remove undesirable cell types that could interfere with T-cell activation and proliferation [176]. The CAR gene constructs expressing a specific antigen are subsequently introduced into T cells through one of the numerous techniques discussed elsewhere [177,178]. Eventually, the CAR T cells are amplified ex vivo to generate an adequate number of cells for the therapeutic dose. When CAR T cells are re-introduced into the patient, the receptors assist the T cells in recognizing cancer cell antigens and the destruction of cancer cells (Figure 3B). Israeli immunologists Zelig Eshhar and Gideon Gross constructed the first engineered T-cell with a chimeric molecule between 1989 and 1993 [179,180]. Subsequently, in 2011, Carl June and David Porter performed the first clinical use of CAR-T cells on patients with chronic lymphocytic leukemia [181,182]. CARs consist of an extracellular ligand-binding domain, typically a single chain variable fragment (scFv), a spacer domain, a transmembrane/hinge domain, and cytoplasmic domains (Figure 3C). The scFv domain is involved in antigen recognition, while the intracellular cytoplasmic domains play a critical role in ligand-dependent signaling [183]. First-generation CARs contained a single activation domain, which typically is the cytoplasmic domain CD3ζ. Second-generation CARs contained an activator domain (CD3ζ/γ chain of Fc receptors) linked to co-stimulatory domains from CD28 and 4–1BB [183,184,185]. Further advancement in CAR T technology has led to the development of third and fourth-generation CAR modules with additional regulatory and co-regulatory functional domains [183,184,185,186,187]. The design of each component of the CAR structure plays an important role in CAR-T effector functions, efficacy, and toxicity [183,184,185,186,187]. With second and third-generation CARs, significant improvement in antitumor efficacy has been achieved in terms of increased T-cell proliferation, apoptosis resistance, cytokine release, and in vivo survival of CART cells [188,189]. TRUCKs (CAR redirected T cells that transport a transgenic product to the targeted tumor tissue) are fourth-generation CARs armed with a transgenic’ payload [188,189]. Fourth-generation CAR T cells can be a promising delivering drug for modifying the tumor microenvironment by the inducible release of transgenic immune modulators. Another engineered version of CAR-T cells, called “smart T cells”, is claimed to be both safer and more efficacious [190].

Since 2017, the U.S. FDA has approved six CAR T cell therapies. The first CART cell therapy to get FDA approval was Tisaglecleucel (Kymriah) for treating B-cell acute lymphoblastic leukemia [191]. Kymriah, a CD19-directed autologous CAR T cell therapy, is approved for resistant B-cell acute lymphoblastic leukemia (ALL) or in the second or later relapse stage. Recently, in May 2022, Kymriah also got accelerated approval for adult patients with relapsed or refractory follicular lymphoma [192]. Five other CAR-T drugs Lisocabtagene maraleucel (Breyanzi), Brexucabtagene autoleucel (Tecartus), Axicabtagene ciloleucel (Yescarta), Idecabtagene vicleucel (Abecma), and Ciltacabtagene autoleucel (Carvykti) have been approved by the U.S. FDA. The active adoptive cell therapies in various stages of clinical development and the timeline of FDA approvals of CART cell therapies are shown in Figure 3D,E respectively. Some of the ongoing clinical trials of CART therapy in various cancers are shown in Table 2.

### 4.4. CAR-NK Cells

In the field of cell-based immunotherapies, CAR NK cells have attracted significant attention, like CAR T cells [193]. CAR NK cells have several advantages over CAR T cells, such as antigen-independent spontaneous cytotoxicity, differentiated cytokine secretion, and better survival in vivo [193,194]. Engineered primary human NK cells and NK-92 cells expressing CAR targeted against CD19, CD20, CD244, and HER2 have been studied in a wide range of preclinical settings. Natural killer (NK) cell therapy, CYNK-001, produced from cryopreserved human placental hematopoietic stem cells, recently received fast-track designation from the U.S. FDA for acute myeloid leukemia (AML) [195]. A phase I/II trial of CD19-targeted CAR-NK treatment obtained from cord blood is now underway in patients with relapsed or refractory CD19+ malignancies (NCT03056339). Having established success in hematological malignancies, many NK cell-based therapies are currently in clinical trials for solid tumors [196,197,198].

Further advancement in NK cell therapy is NK-cell engagers, which aid in the identification of target cells as well as NK cell activation. This approach is called ANKET—antibody-based NK-cell engager therapy [199,200,201]. The first generation ANKETs are antibody-like molecules that target two NK cell activation receptors, CD16 and NKp46a, as well as a tumor-specific antigen. When these engager antibodies bind to NK cell receptors, they connect NK cells to target tumor cells and activate the NK cell’s effector function [196,199]. The first ANKET therapy, SAR443579, which targets the tumor antigen CD123 [202], is currently being tested in patients with relapsed or refractory acute myeloid leukemia (R/R AML), B-cell acute lymphoblastic leukemia (B-ALL), or high-risk myelodysplasia (HR-MDS) (NCT05086315). Although CAR NK therapy seems promising, some of the challenges that need to be addressed in future studies include acquiring a significant number of NK cells to achieve a therapeutic scale and sustaining *in vivo* survival and activity after administration [193,196,197,203].

### 4.5. Limitations and Challenges of CAR T Therapy

CAR T cell therapy has several limitations, which include antigen escape, antigen heterogeneity, trafficking of CAR T cells and tumor infiltration, immunosuppressive microenvironment, and CAR-T cell-associated toxicities [204]. One of the most challenging limitations of CAR-T cell therapy is the emergence of tumor resistance to single antigen targeting CAR constructs. Despite the ability of single antigen CAR-T cells to induce a potent response, some malignant cells in patients can exhibit a partial or complete loss of target antigen expression. This mechanism is known as antigen escape [204,205]. Recent follow-up data from relapsed and/or refractory ALL patients and multiple myeloma treated with CD19 CAR-T therapy or B-cell maturation antigens (BCMA) targeted CAR-T cells indicates that the incidence of resistance to therapy in a small percentage of patients is due to the loss of CD19 and BCMA [204,205,206,207,208]. Although CAR-T therapies have shown great potential in treating hematological cancers, their efficacy remains undetermined in solid tumors. Targeting solid tumor antigens is challenging since many of these antigens are often expressed to varying extents by normal tissues. Antigen selection is, therefore, critical for CAR-T cell design to enhance therapeutic effectiveness and reduce off-target effects. [204]. There has been considerable effort and an ongoing research area to identify and characterize tumor-specific antigens in solid tumors. Previous studies have investigated pan tumor antigens such as B7-H3 [209] and Muc16 (ecto) [210]. A few unrelated studies have focused on targeting tumor-specific post-translational modifications, which are predominant in solid tumors [211]. Another factor that limits CAR-T cell therapy’s effectiveness in solid tumors is the restricted ability of CAR-T cells to infiltrate. The physical barrier and immunosuppressive microenvironment restrict the penetration and mobility of CAR-T cells to the target site [204]. One of the approaches to overcome these limitations is using modified delivery methods that (1) eliminate the requirement for CAR-T cell trafficking to target sites and (2) minimize interaction with the normal tissues and limit on-target off-tumor toxicities of CAR-T cells [204]. One study demonstrated that expressing chemokine receptors on CAR-T cells can significantly improve CAR-T cell trafficking in response to tumor-derived chemokines [212]. The immunosuppressive environment created due to T cell exhaustion and low *in vivo* persistence limits the efficacy of CAR-T responses. It is hypothesized that T cell exhaustion occurs due to the activation of co-inhibitory pathways [213]. Therefore, combining immunotherapy with checkpoint inhibitors and CAR-T cell therapy is presumed to be efficacious for robust T cell effector function in solid tumors and hematological cancers [204,213,214]. Although CAR-T cell therapy has been a potential cancer therapeutic strategy, its capability to become a first-line treatment has been constrained by frequent toxicities such as cytokine release syndrome (CRS) [132].

The efficacious therapeutic response of CAR- T therapy is determined by the extent of CAR-T cell activation and cytokine secretion after engaging with the target antigen. The CAR-T cell activation is largely influenced by factors such as the level of tumor antigen, tumor burden, the affinity of the antigen binding domain to its target epitope, and the CAR’s costimulatory elements [215,216]. Future modified approaches may include refining CAR components, their modular structure, and activation kinetics to optimize therapeutic potential and reduce toxicity. Another potential strategy for reducing CAR-T cell-mediated toxicity would be to incorporate bimodal “off-switches” that would allow modified cells to be selectively deactivated at the onset of adverse events [204,215,216]. Strategies using tandem CARs or dual CARs that simultaneously recognize two or more different tumor antigens in a single design with two scFvs have reduced the relapse rate in CAR T treatments [217]. Preliminary findings from clinical trials involving dual-targeted CAR-T cells have been promising [218]. Given the potential of adoptive cell therapies to translate into effective anticancer treatment, further advancements in the field will lead to more promising and reliable tools.

## 5. Monoclonal Antibodies

A potential class of targeted anticancer therapeutics with a variety of mechanism(s) of action are monoclonal antibodies (mAbs) [219,220,221,222]. The U.S. FDA has approved more than 100 monoclonal antibodies (mAbs) to treat a variety of human disorders, including cancer and autoimmune and chronic inflammatory diseases [223,224]. Since the successful applications of immunoglobulin G (IgG) mAbs, there have been significant advances in antibody engineering technology that led to the development of newer and more efficacious antibody formats and derivatives such as antibody fragments, non-IgG scaffold proteins, bispecific antibodies (BsAbs), antibody-drug conjugates (ADCs), antibody-radio conjugates, and immunocytokines that have successively been used as alternative therapeutic agents for a broad range of cancers [219,220,221,222,225,226]. Monoclonal antibodies attack cancer cells in various ways (Figure 4A), including directly targeted killing, immune-mediated destruction, blocking immune checkpoints, preventing blood vessel formation, and delivering a cytotoxic drug to cancer cells [219,220,221,222,225,226].

### 5.1. Direct Killing and Immune-Mediated Killing

MAbs, upon antigen-specific binding to tumor-associated antigens, induce cytotoxic effects either by neutralizing or killing through cell-intrinsic proapoptotic signaling mechanisms [219,222,224]. One example is Trastuzumab, an anti-HER2 monoclonal antibody that inhibits cancer cell growth by interfering with HER2 dimerization and intracellular signaling [219,220]. The immune-mediated mechanism involves antibody-dependent cellular cytotoxicity (ADCC), complement-dependent cytotoxicity (CDC), and antibody-dependent cellular phagocytosis (ADCP). Several FDA-approved monoclonal antibodies, targeting CD20 (Rituximab, Ofatumumab, Obinutuzumab), CD38 (Daratumumab, Isatuximab), CD52 (Alemtuzumab), SLAMF7 (Elotuzumab), HER2 (Trastuzumab, Pertuzumab), EGFR (Cetuximab, Panitumumab, Necitumumab), GD2 (Dinutuximab) and CCR4 (Mogamulizumab) for hematological and solid malignancies work utilizing one or more mechanisms among CDC, ADCP, and ADCC. [220,221,226,227]. Moreover, many other mAbs, Margetuximab (anti-HER2), Tomuzotuximab (anti-EGFR), Ublituximab (anti-CD20), Gatipotuzumab (anti-MUC1), MOR208 (anti-CD19) and MEDI-551 (anti-CD19) are in later stages of clinical trials [221,227,228,229].

### 5.2. mAbs Targeting Angiogenesis

Angiogenesis, or the formation of new blood vessels, is vital for tumor growth and progression [230]. Inhibiting angiogenesis has long been seen as a promising avenue for the generation of new, effective, and targeted cancer treatments capable of inhibiting tumor growth and spread [230,231,232]. The use of monoclonal antibodies that target the vascular endothelial growth factor (VEGF) pathway has significantly improved cancer treatment [232]. The U.S. FDA has approved a variety of angiogenesis inhibitors for the treatment of solid cancers. Bevacizumab, a humanized monoclonal antibody that targets VEGF, is approved as monotherapy or in combination with other drugs to treat a range of malignancies, including colorectal cancer, recurring glioblastoma, metastatic and irresectable hepatocellular carcinoma, non-squamous non-small cell lung cancer (NSCLC), ovarian and renal cell carcinomas [231,233,234,235,236,237,238]. Another monoclonal antibody, Ramucirumab, targeting VEGFR2, is currently being used for advanced gastric cancer, non-small cell lung cancer, metastatic colorectal cancer, and hepatocellular carcinoma (HCC) [233,239,240,241].

### 5.3. Antibody-Drug Conjugates

ADCs are the most rapidly emerging monoclonal antibody-based cancer immunotherapy field, with significantly greater potency than naked antibodies [242,243]. This strategy employs a mAb attached to the cytotoxic payload via a chemical linker directed to a target cell surface antigen expressed on the cancer cells [244,245,246]. Typically, ADC contains three components: the monoclonal antibody (mAb), the linker, and the cytotoxic or cytostatic/cytotoxic drug. The mAb is an essential component of ADC that dictates specificity to the tumor cell, by binding to tumor antigens, acts as a precise carrier to transport the cytotoxic agents to the target cell without harming healthy cells that do not express the target antigen [242,243,245,246,247,248]. The second component of ADC is a linker. a chemical entity that links the mAb to the cytotoxic drug [247,249,250]. The two types of linkers are used based on the drug release mechanisms, (1) Cleavable linkers; that are cleaved in the presence of low lysosomal pH or certain enzymes found in different cellular compartments; cytotoxic substances are released, (2) non-cleavable linkers, on the other hand, require ADC proteolysis to release the active cytotoxic agent [247,249,250,251]. The cytotoxic agent is the third important component of ADC also known as ‘payload,’ which typically is a cytotoxic drug such as microtubule-disrupting agents (maytansinoids, auristatin analogs, and tubulysins) or DNA-damaging agents (duocarmycins, calicheamicins, and pyrrolo-benzodiazepines) [246,250,251,252].

ADCs are complex structures that require careful consideration of their many components. The choice of a suitable target, the specificity of mAb, the cytotoxic payload, and the method by which the antibody is coupled to the payload are all important factors that may determine the safety and efficacy of ADCs [243,245,250,252,253]. The specificity of the antibody to the target antigen limits systemic exposure and toxicity [245]. For an efficient ADC, a high antigen expression level is crucial for ADC success. In addition, internalization of the target antigen following interaction with ADCs is critical to delivering the cytotoxic agent to the target site in cells [245,246,250,251]. Secondly, the linker’s choice and stability in blood circulation play a crucial role in avoiding unwanted effects on healthy cells [246,250,254]. Third, the cytotoxic payload should be of high potency (IC_50_ in the nanomolar and picomolar range) [254]. Gemtuzumab ozogamicin (Mylotarg), a recombinant humanized immunoglobulin G4 (IgG4) mAb directed against CD33 with a pH-sensitive hydrazone linker and a calicheamicin derivative (N-acetyl-gamma-calicheamicin-dimethyl hydrazide) payload, was the first ADC to receive FDA approval in 2000 for the treatment of relapsed Acute myeloid leukemia [255]. Since then, more than 14 ADCs have received FDA approval (Figure 4B), and more than 100 ADCs are in various phases of clinical trials [252,254]. The approved ADCs include: Adcetris (brentuximab vedotin); Kadcyla (ado-trastuzumab emtansine); Besponsa (inotuzumab ozogamicin); Polivy (polatuzumab vedotin); Padcev (enfortumab vedotin); Enhertu (fam-trastuzumab deruxtecan); Trodelvy (sacituzumab govitecan; Blenrep (belantamab mafodotin); Zynlonta (loncastuximab tesirine); Tivdak (tisotumab vedotin). A recent systematic review by Fu Z et al. has given a detailed view of ADC generations, payloads, and clinical developments so far [254].

Recently in 2022, Fam-trastuzumab deruxtecan-nxki (Enhertu, Daiichi Sankyo, Inc.), an anti-HER2 ADC, received FDA approval for the treatment of metastatic and unresectable HER-2 low breast cancer. Enhertu is the first targeted therapy approved for HER-2 low subtype of breast cancers. The DESTINY-Breast04 clinical study data, which included 557 patients with metastatic breast cancer, served as the ground for the approval (NCT03734029). The median progression-free survival (PFS) for patients who received Enhertu was 9.9 months, compared to 5.1 months for patients who received chemotherapy (www.fda.gov, accessed on 7 August 2022).

### 5.4. Antibody Radioimmunoconjugate (RIC)

A monoclonal antibody (mAb) coupled to a radionuclide is called radioimmunoconjugate [256,257], RICs. Unlike ADCs, they do not require cellular uptake or endocytosis to provide anti-cancer activity; instead, DNA strand breaks occur in the target cell when a mAb emitting radionuclide attaches to its targeted antigen [256,257,258]. The advantage of radioimmunotherapy (RIT) over standard radiotherapy is that it is less hazardous and improves the efficacy of mAbs [256]. Radio-immunoconjugates (RICs) have been successfully developed as theragnostic tools, with the FDA approving Ibritumomab tiuxetan (Zevalin^®^; Biogen Idec) and ^131^I-tositumomab radioimmunoconjugate for the treatment of non- Hodgkin’s lymphoma. Ibritumomab tiuxetan is a yttrium-90, and Tositumomab is an iodine-131 radionuclide coupled monoclonal antibody that targets the CD20 antigen on B-cell malignancies [256,257,258,259,260]. Even though Ibritumomab and Tositumomab are the only FDA-approved RIT, many RICs have entered clinical studies for both hematological and solid cancers. Other than the therapeutic usage, RICs have a wide range of applications in cancer imaging and diagnosis [256,257,261]. Currently, 37 RICs are under phase I/II/III clinical trials for the diagnosis or treatment of various human cancers [256,257].

### 5.5. Bispecific Antibodies

Bispecific antibodies (BsAbs), a complex family of antibodies, can bind two distinct antigens epitopes [262]. Bispecific antibodies (BsAbs) have received a lot of interest as a new generation approach of monoclonal antibody-target cancer immunotherapy that focuses on engaging immune cells with cancer cells or concurrently targeting two receptors [262,263,264]. Bispecific T cell engagers (BiTEs) are essentially bispecific antibodies with two variable fragments of single-chain antibodies. One of the bispecific antibodies targets a cell surface molecule on T cells, such as CD3, while the other specifically targets the surface antigens on tumor cells. The immunological synapse created by contacts between immune cells and tumor cells via BiTEs results in the production of effector cytokines and the killing of the tumor cell [262,263,264,265]. Currently, the FDA has approved four bispecific antibodies, Blinatumomab, Emicizumab, Amivantamab, and Faricimab-Svoa, while three others, Emicizumab, Amivantamab, and Mosunetuzumab, have been approved by European Medicine Agency (EMA) [262,263,264,265,266,267,268,269].

## 6. Cytokine Therapies

Cytokines are immune system components that play an important role in the cancer immunity cycle; however, the expression and activity of many cytokines are dysregulated in malignancies [270,271,272]. The first cytokine to be used in the treatment of cancer was IL-2. It is considered not only the first cytokine therapy but also the first reproducible and effective human cancer immunotherapy [273].

The U.S. FDA approved IL-2 in 1992 for the treatment of metastatic renal cell carcinoma, and in 1998, subsequently, it was later approved for metastatic melanoma [274]. Though high-dose IL-2 monotherapy demonstrated encouraging outcomes in metastatic renal cell carcinoma and melanoma, its use remained limited due to toxicity and high production costs [273,274]. Another challenge with IL-2 is that it can activate both cancer-killing effector T cells as well as immunosuppressive regulatory T cells. Therefore, further research is required to better understand the complex biology of IL-2 to harness its utility as cancer immunotherapy. Moreover, despite significant preclinical data, no other cytokines nor cytokine antagonists have been successful as monotherapies in advanced-stage cancer patients. However, the FDA approved a few cytokine therapies as adjuvant therapies, including interferon-alpha and its pegylated form, PEG-interferon-2b (PEG-IFN), for resected stage III melanomas [275]. BEMPEG (Bempegaldesleukin), a pegylated version of IL2, had advanced to the third clinical trial stage in combination with Nivolumab for metastatic melanoma (NCT03435640). However, this combination therapy showed no significant improvement in the overall survival of melanoma patients when compared to Nivolumab monotherapy (NCT03435640). Moreover, in patients with advanced clear-cell renal cell carcinoma, BEMPEG, combined with Nivolumab, has shown promising results in Phase 1b/II clinical trials (NCT02989714). Additionally, it is currently being tested in phase II/III trial for metastatic and recurrent head and neck cancer (HNSCC) and phase I/II for non-small cell lung cancer (NSCLC) with another PD1 inhibitor, Pembrolizumab (NCT02989714, NCT03138889). Several other cytokines, including (IL-8, IL-10, IL-12, IL-15, TNFα, TGFβ, CSF1, CCL2/3/5, and VEGF) are in preclinical or different clinical developmental stages in combination with other cancer immunotherapies such as immune checkpoint inhibitors, CART cells, TILs, NK cells and chemotherapy [270,272,276,277,278,279]. Hence, the application of cytokines is not restricted to direct usage as a therapeutic agent, but they also play a critical role in developing cell-based therapies. In fact, the adoptive T cell therapies totally rely on cytokines for *in vitro* proliferation and *in vivo* persistence [270,272,276,279].

### Limitations and Challenges of Cytokine Therapy

One of the fundamental limitations of cytokine therapy is the pleiotropic effect of cytokines [270,280]. Each cytokine can affect numerous cell types that elicit both pro- and anti-inflammatory responses [270,276]. The expensive cost of manufacture, the need for producing clinically required doses to elicit a reliable immunological response, as well as the short half-life and systemic toxicity, are further barriers to the effectiveness of cytokine therapy [270,276,277]. To address these constraints, further research is required to develop optimized formulations of cytokines, improved methods of delivery, and their applicability in combinatorial immunotherapy approaches.

## 7. Oncolytic Viruses

Oncolytic viruses (OVs) are genetically modified viruses that lack virulence but can still attack and kill cancer cells without harming healthy cells [281]. OVs are a unique class of therapeutic agents that exhibit broad-spectrum activity to induce tumor cell death and enhance both innate and tumor-specific adaptive immune responses [281,282]. Oncolytic viruses can kill cancer cells in various ways, including direct virus-mediated cytotoxicity and cytotoxic immune effector mechanisms [281]. A model depicting typical OV and its mode of action is shown in Figure 5A.

OVs are superior to other immunotherapy methods due to their exceptional capacity to target tumors without relying on antigen expression patterns. OVs can reprogram the immune suppressive tumor microenvironment (TME), promote the recruitment of tumor-infiltrating lymphocytes, and enhance robust antitumor immunity [283]. To date, three OVs have been approved for cancer treatment. Rigvir, an RNA virus derived from the native ECHO-7 picornavirus strain, was approved in 2004 in Latvia to treat melanomas [284]. In 2005, an engineered adenovirus, H101, was approved in China for nasopharyngeal carcinomas [285]. In 2015, the U.S FDA approved Talimogene laherparepvec (T-VEC), an attenuated Herpes simplex virus (HSV-1 encoding GM-CSF, for recurrent melanomas [286]. Oncolytic viral therapy is now an accepted form of immunotherapy with the development of T-Vec. Several oncolytic viruses (OVs), such as HF10 (Canerpaturev—C-REV) and CVA21 (CAVATAK), are now being studied in phase II/III as monotherapies or in combination with immune checkpoint inhibitors against melanoma (NCT01227551). Another OV, Pexa-Vec, is undergoing phase I/II clinical trials for refractory colorectal cancer in combination with immune anti-CTLA4 checkpoint inhibitors (NCT03206073). Pexa-Vec (JX-594) is an oncolytic vaccinia virus engineered to generate β-galactosidase and human GM-CSF transgenes. In addition to their direct oncolytic activities, Pexa-Vec has been shown to increase tumor cell death through activating innate and adaptive immune responses [287]. Provisional regulatory approval for an HSV-based OV, Delytact (Teserpaturev/G47Δ), by Daiichi Sankyo in Japan for intractable gliomas, is the most recent advancement in the field of OV therapy [288]. G47∆ is a genetically engineered third-generation oncolytic HSV-1, herpes simplex virus type 1 (HSV-1), with triple mutations within the viral genome [289,290]. G47 is effective through two different mechanisms: (1) an immediate effect through selectively enhanced virus replication in tumor cells and direct oncolytic action; and (2) a delayed impact through the establishment of targeted antitumor immunity. G47Δ shows enhanced cytopathic effects while retaining a high safety profile [289]. A phase 2 single-arm trial evaluating the efficacy of G47∆ observed specific antitumor immune responses, such as an increased tumor-infiltrating CD4^+^ and CD8^+^ lymphocyte populations after intratumoral administration of G47∆ in glioma patients [289]. To date, 13 different viruses have been studied in preclinical and clinical studies, with the majority focusing on oncolytic adenoviruses, HSV, and the vaccinia virus [281], with over 100 clinical trials now underway [291].

### Limitations and Challenges of OV Therapy

Despite their numerous potential therapeutic benefits, oncolytic viruses (OVs) are not widely utilized as an effective anticancer treatment modality. Many factors limit OV-based immunotherapies, such as viral tropism, delivery platform, antiviral immunity, tumor heterogeneity, tumor microenvironment, and oncolysis by OV [281]. Despite adequate preclinical evidence of OV’s antitumor efficacy and reasonable safety profile, its applications in clinical settings have been slow [291].

The existing challenges in OV therapy include inadequate understanding of tumor suppression mechanism(s) with a specific oncolytic virus agent, lack of specific biomarkers to connect potent viral species with receptive tumor types and patient attributes, and poor uniformity of immune correlates in clinical trials [291]. However, choosing an optimal OV species, efficient delivery platforms, and retargeting OVs using genome engineering for better penetrance in solid tumors are some of the strategies that have been proposed to reinforce OVs in therapeutic settings [281].

## 8. Cancer Vaccines

Recent advances in our understanding of various molecular mechanisms that tumor cells utilize to evade immune surveillance have facilitated the development of preventative and therapeutic cancer vaccines that could be effective against a wide range of human malignancies [292]. Cancer vaccines stimulate the immune system to mount an antitumoral response and protect against cancer [124,293]. Cancer vaccines are typically classified as either prophylactic or preventative or therapeutic. Prophylactic vaccines protect against oncogenic virus infection. Therapeutic vaccinations, on the other hand, harness the potential of the immune system to eradicate neoplastic cells. [292,293].

Prophylactic vaccines against the hepatitis B virus (HBV) and the human papillomavirus (HPV) have been effective in lowering the incidence of hepatocellular carcinoma and cervical cancer, respectively [292]. The prophylactic vaccines for HBV and HPV have been approved by U.S. FDA. Following its approval in 1981, the anti-HBV vaccine was the first preventive cancer vaccine used in clinical settings [292]. It has been demonstrated that the HBV vaccine comprising recombinant HBV surface antigen (HBsAg) confers lifetime immunity [294]. There has been a significant decline in hepatitis-associated hepatocellular carcinomas globally following the HBV vaccination programs [292,295]. HPV vaccines approved by the US FDA are Gardasil and Cervarix, derived from viral subunit-like particles (VLPs) and confer protection against HPV (types 16 and 18) [296]. Cervarix is a bivalent vaccine, Gardasil is a quadrivalent vaccine, and Gardasil-9 is a 9-valent vaccine [297].

Therapeutic cancer vaccines involve the exogenous administration of specific tumor antigens to the patient to activate their adaptive immune system and elicit an anti-tumor response [293] (Figure 6). Bacillus Calmette–Guérin (BCG), often used as a preventative tuberculosis vaccine, was repurposed in 1990 as the first immunotherapy approved by the FDA for early-stage bladder cancer [298]. After that, more than two decades of preclinical and clinical research on therapeutic cancer vaccines has yielded only one therapeutic cancer vaccine, Sipuleucel-T, approved by the U.S FDA for symptomatic metastatic castration-resistant prostate cancer (mCRPC) [233,299]. Sipuleucel-T is a cell-autologous vaccine made by enriching patients’ dendritic cells (DC) and stimulating them with PAP immunogen and GM-CSF. The Sipuleucel-T treatment produces PAP-specific T cells capable of identifying and killing PAP-expressing prostate cancer cells [299,300]. A pivotal phase III clinical trial (IMPACT) of Sipuleucel-T in patients with mCRPC showed a slight but significant improvement in median overall survival (OS) [301]. However, the complexity and high expense of producing Sipuleucel-T have limited its widespread use [257].

Typically, therapeutic cancer vaccines target two classes of antigens, tumor-associated antigens (TAAs) and tumor-specific antigens (TSAs). TAAs are self-antigens that may be expressed to some degree in a subset of normal cells but are either abnormally or preferentially expressed on tumor cells. TSAs are tumor-specific, arising from oncogenic viruses or oncogenic driver mutations that generate neoantigens [302]. Therapeutic cancer vaccines are divided into four categories depending on the various formulation methods and delivery systems: nucleic acids (DNA or RNA)-based vaccines, viruses-based vaccines, peptide-based vaccines, and cell-based vaccines [292]. Vaccines that utilize whole cells such as autologous antigen-presenting cells such as dendritic cells (DCs) as antigen carriers are called cell-based cancer vaccines. Peptide-based vaccines comprise predicted epitopes of tumor antigen. Virus-based vaccines utilize viral vectors expressing the target tumor antigen. DNA vaccines may encode TAA or immunomodulatory factors to induce tumor antigen-specific response. mRNA cancer vaccine formulations comprise *in vitro* synthetic mRNA encoding a single or multiple antigens. Each of these has some advantage over the other, reviewed elsewhere [292,303].

The last decade has seen significant progress and technological advancements in cancer vaccine research, such as tools for immunogenomic profiling and the discovery of novel antigen repertoires, formulation methodologies, and delivery platforms [304]. Advances in cancer vaccine research have resulted in the development of many therapeutic cancer vaccine candidates that effectively increase antigen presentation, activate effector T cells, and overcome tumor-induced immunosuppressive pathways. Many of these are in various preclinical and clinical trial stages [302,303,304]. Table 3 lists the selected ongoing cancer vaccine clinical trials.

### Limitations and Challenges of Cancer Vaccines

Although cancer vaccines have shown tremendous preclinical promise, many fail in therapeutic settings. There have been several setbacks in therapeutic vaccine development [292,300]. One of the most critical constraints has been antigen selection [305]. Although the concept of neoantigens has emerged, predicting which neoantigens can elicit a robust antitumor response remains a daunting task [302]. Cancer vaccines have been unsuccessful in patients with ‘cold tumors’ which are refractory to immunotherapy [18]. ‘Cold tumors’ are typically characterized by a low infiltration of effector T cells in the immunosuppressive tumor microenvironment, a low mutational load, and a low neoantigen burden. [227]. Many cancer vaccines have failed to elicit clinical efficacy, partly due to the tumor’s immune evasion and escape mechanisms such as loss of antigenicity, loss of MHC-I, presence of an immune suppressive tumor microenvironment, and a paucity of a strong antitumor immune response [226].

## 9. Conclusions and Future Perspectives

Cancer immunotherapies, such as ICIs and CAR-T, have dramatically altered the treatment landscape for many solid and hematologic malignancies [124,187]. Ongoing research continues to investigate ways to harness the immune system to treat cancer and broaden the indications for currently available therapies. Although immunotherapies have revolutionized the treatment of solid and hematologic malignancies, they have unique toxicity profiles based on their mode of action [131,132]. Despite this, such innovative therapies can potentially increase already-in-use therapies’ effectiveness. More preclinical research is warranted to improve the side effects and therapeutic responsiveness of immune checkpoint inhibition by altering existing antibodies or developing combinatorial methods with immunomodulators. Recent research has led to the invention of innovative delivery systems and modifying existing antibodies to mitigate side-effect and enhance the therapeutic efficacy of immune checkpoints [15,131]. Numerous clinical trials are ongoing to assess ICI’s safety and efficacy in conjunction with traditional cancer therapy targeted molecular drugs and novel immunomodulatory medicines. Moreover, newly identified innate immune checkpoints such as CD47 and CD24 have opened new avenues of cancer treatment and have demonstrated preclinical success. Harnessing the phagocytic potential of macrophages represents a promising approach. However, more rigorous preclinical studies followed by their applicability in clinical trials are required to establish CD47 and C24 as clinically targetable innate immune checkpoints.

Further research is needed in CAR-T cell-based therapies, particularly for solid tumors, to determine how to prevent antigen escape and select highly immunogenic tumor-specific antigens that elicit a potent antitumor immune response while limiting toxicity profile. Future research must focus on improving tumor penetration of CAR T cells to extend therapeutic efficacy to solid tumors. To reduce the toxicities associated with checkpoint therapy or CAR T therapy and immune-related adverse events, metabolic and pharmacogenomic profiling of patients may provide greater insight into the essential genes and pathways that mediate toxicity or the adverse effects. Several recent studies have demonstrated the influence of the gut microbiota, specifically on the response of immune checkpoint inhibition across cancer types [306,307]. Several ongoing clinical trials investigate the therapeutic potential of manipulating gut microbiota directly in cancer patients [306]. Future gut microbiome research may unravel more therapeutic opportunities in conjunction with immunotherapy. Therapeutic cancer vaccines have increased response rates and survival, especially with ICI. The discovery of somatic mutations and the study of peptides derived from these alterations to trigger immune responses has reignited interest in the therapeutic cancer vaccine. Much effort is required to identify neoantigens, create combination therapies, and optimize vaccination platforms before cancer vaccines become viable immunotherapy. Therapeutic cancer vaccines offer the opportunity to develop personalized immunotherapies and boost immune memory. Clinical trials for vaccines utilizing a wide variety of cancer-specific neoantigens with high patient-specificity have had encouraging results thus far [137,305]. Combining checkpoint inhibitors with tailored cancer vaccines and innovative targeted therapeutics may be the future of cancer immunotherapy. For oncolytic virus-based therapies, an in-depth understanding of potential challenges and continued investigation of recombinant oncolytic virus subtypes should build a platform for their broad applicability in cancer treatment.

Given that ongoing preclinical research continues to investigate more effective immunotherapeutic tools to eradicate cancer, more efforts are needed to address pitfalls and toxicities associated with these therapies simultaneously. Identifying the mechanisms and pathophysiology leading to toxicity will improve current immunotherapy approaches. Bringing clinical advantages to patients necessitates a thorough understanding of the mechanisms underlying a specific and effective antitumor response and limiting the toxicity associated with various immunotherapy approaches. As clinical trials continue to expand for various immunotherapy-based therapeutics, a broader application of these promising therapies, from solid tumors to complex hematological malignancies, requires the development of appropriate risk-based stratification models. Despite existing limitations, modified strategies and emerging innovative solutions may lead to the future development of more effective and safe cancer immunotherapeutics.

## Figures and Tables

**Figure 1 diseases-10-00060-f001:**
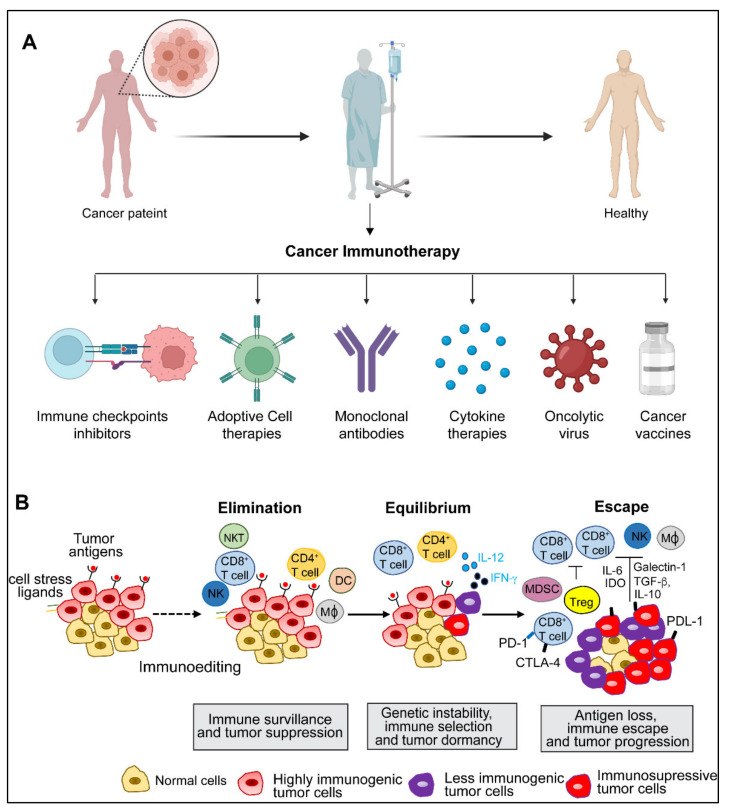
(**A**) Illustration depicting various immunotherapy approaches for cancer treatment. Created with BioRender.com. (**B**). Three steps of cancer immunoediting process: elimination, equilibrium, and escape. (Abbreviations: NK, natural killer; NKT, natural killer T; DC, dendritic cell; MΦ, macrophage; MDSC, myeloid-derived suppressor cells, IFN, interferon; IL, interleukin; CTLA-4, cytotoxic T lymphocyte-associated protein-4; PDL-1, programmed cell death ligand 1; PD-1, programmed death 1; Treg, T regulatory; IDO, indoleamine 2,3-deoxygenase; TGF-β, transforming growth factor-β).

**Figure 2 diseases-10-00060-f002:**
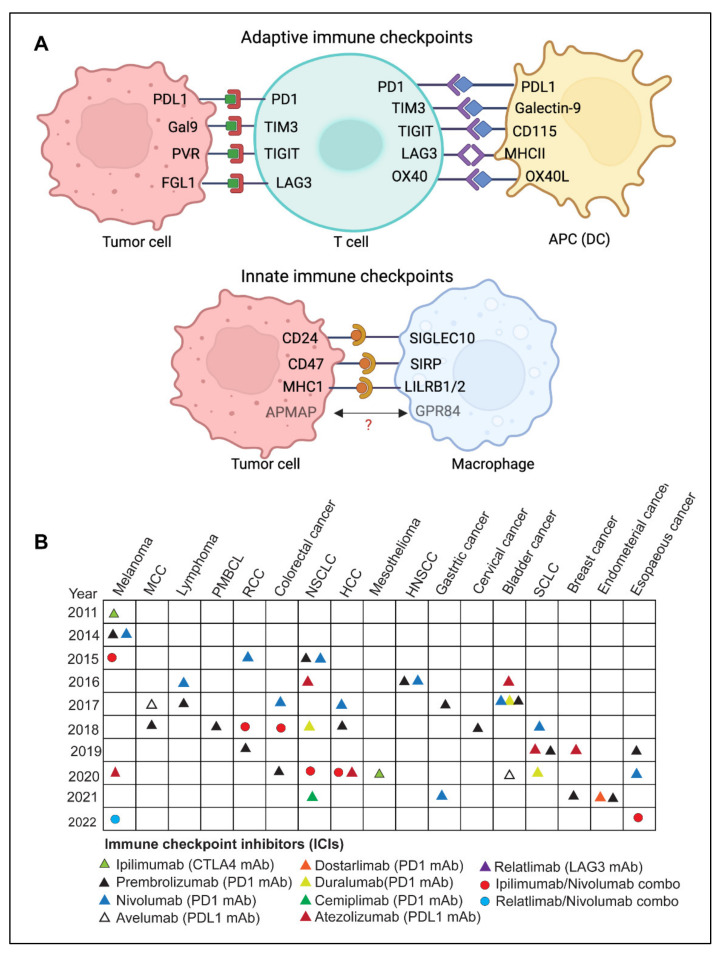
(**A**) Overview of adaptive and innate immune checkpoint molecules involved in the antitumor response, present on T cells and antigen-presenting cells, such as dendritic cells and macrophages. Created with BioRender.com. (Abbreviations: PD-1, programmed death-1; PDL-1, programmed cell death ligand 1; TIM-3, T cell immunoglobulin and mucin domain-containing protein 3; Gal-9, galectin 9; TIGIT, T cell immunoreceptor with immunoglobulin and ITIM domain; PVR, Poliovirus receptor; LAG-3, lymphocyte-activation gene 3; FGL1, fibrinogen-like protein 1; MHC-I, major histocompatibility complex I; MHC-II, major histocompatibility complex II; Siglec-10, sialic acid binding Ig like lectin 10; SIRPα, signal regulatory protein alpha, APMAP, adipocyte plasma membrane-associated protein; GPR84, G-protein coupled receptor 84; LILRB, leukocyte immunoglobulin-like receptor. Figure generated using BioRender.com. (**B**) Timeline of U.S. FDA-approved immune checkpoint inhibitors (ICI) for various cancers. Data source: www.cancerresearch.org, accessed on 25 June 2022. (Abbreviations: MCC, merkel cell carcinoma; PMBCL, primary mediastinal large B-cell lymphoma; RCC, renal cell carcinoma; NSCLC, non-small lung cell carcinoma; HNSCC, head and neck squamous cell carcinoma; SCLC, small-cell lung carcinoma).

**Figure 3 diseases-10-00060-f003:**
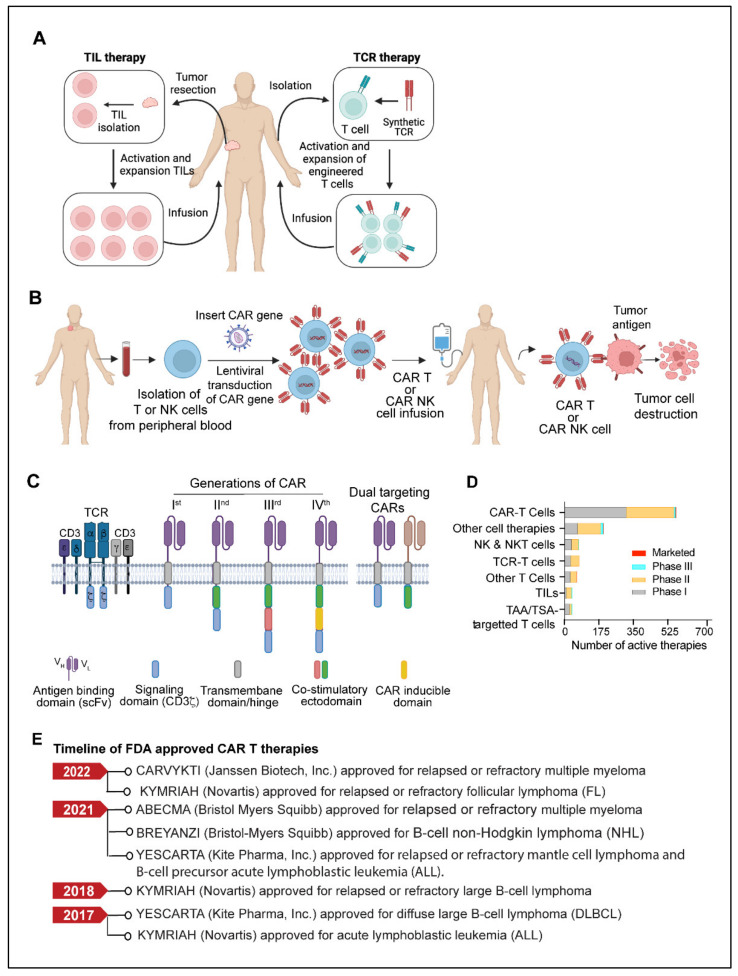
(**A**) Model showing TIL therapy and TCR therapy. (**B**) Schematic outline of CAR T or CAR NK therapy. Briefly, T cells or NK cells are isolated from the apheresis blood of a cancer patient, followed by the lentiviral mediated introduction of the CAR gene. The CAR-expressing cells are expanded ex vivo and infused back into the patient’s body. (**C**) Model depicting CAR structure and generations of different CARs. An antigen-binding domain typically consists of variable heavy (VH) and light (VL) chains from a monoclonal antibody assembled through a linker sequence to form a single chain variable fragment (scFv). In first-generation CAR, the scFv is linked via a hinge and transmembrane domain to the CD3ζ, an intracellular T cell signaling domain of the T cell receptor. In the second and third-generation CARs, an additional one or two co-stimulatory domains are present. The fourth generation CARs are typically equipped with inducible domains A dual-targeting CAR contains two CARs, each targeting an independent antigen. (Created with BioRender.com) (**D**) Plots showing the preclinical and clinical trial status of different cell-based therapies for cancer treatment. (**E**) Timeline of U.S. FDA-approved CAR T therapies for various cancers. (Abbreviations: CAR, chimeric antigen receptor; NK, natural killer; TCR, T cell receptor; NKT, natural killer T cell; TAA, tumor-associated antigen; TSA, tumor-specific antigen; TIL, tumor-infiltrating lymphocytes; FDA, food and drug administration).

**Figure 4 diseases-10-00060-f004:**
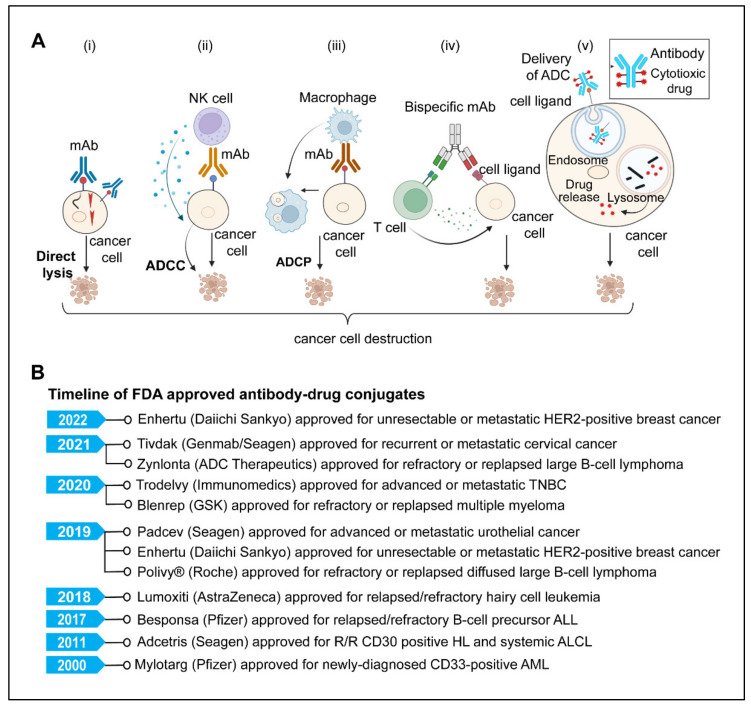
(**A**) Model showing various modes of action of therapeutic monoclonal antibodies used in cancer immunotherapy. (**i**) Direct killing of cancer cells; therapeutic mAbs bind to cancer cell surface receptors and facilitate killing directly by inhibiting downstream signaling pathways that are important for cellular viability and proliferation. (**ii**) Antibody-dependent cell-mediated cytotoxicity (ADCC), an immunological process (primarily mediated by NK cells) in which Fc receptor-bearing effector cells detect and kill antibody-coated target cells expressing tumor antigens on their surface. (**iii**) Antibody-dependent cellular phagocytosis (ADCP); opsonization of cancer cells by monoclonal antibodies triggers Fc receptors on macrophages, resulting in phagocytosis and subsequent killing of internalized target cells by phagosome acidification. (**iv**) A bispecific antibody acts as an adaptor molecule between the effector immune cell and the tumor cell, activation and cross-linking of effector cells (T cells) with tumor cells results in tumor cell lysis (**v**). The ADC (antibody-drug conjugate) binds to the tumor cell’s surface antigen receptor, forming an endocytosed ADC-antigen complex. The internalized complex is then processed by lysosomes, releasing the cytotoxic payload, interacting with its target, and killing the tumor cells. Created with BioRender.com (**B**) Timeline of U.S. FDA-approved antibody-drug conjugates for different cancers (Abbreviations; mAbs, monoclonal antibodies; ADCC, antibody-dependent cell-mediated cytotoxicity; ADCP, antibody-dependent cellular phagocytosis; ADC, antibody-drug conjugates).

**Figure 5 diseases-10-00060-f005:**
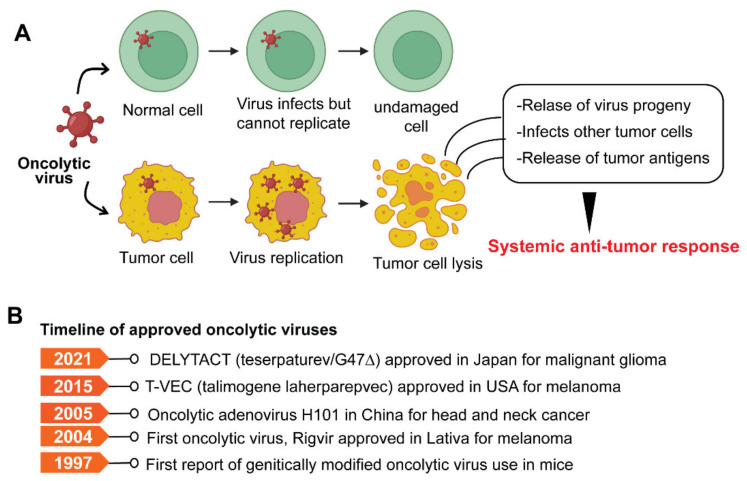
(**A**) The model illustrates the mechanism of oncolytic viruses and the selective killing of tumor cells. (**B**) Timeline of approved oncolytic virus-based therapies.

**Figure 6 diseases-10-00060-f006:**
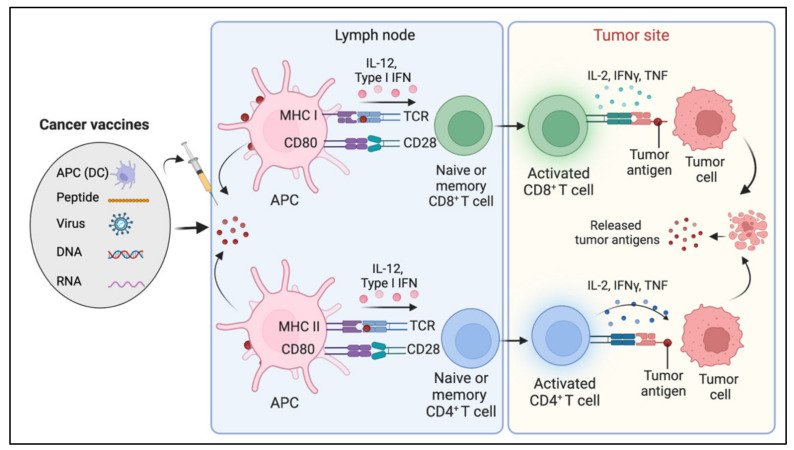
The schematic outline of the mode of action of cancer vaccines. After the administration of a cancer vaccine, APCs such as dendritic cells (DCs) update the antigens, process them, and present them on MHC-I and MHC-II molecules. Antigen-loaded DCs then transmits the signal to cytotoxic T cells through MHC-I and helper T cells through MHC-II, leading to their activation. Upon activation, antigen-specific CD4^+^ or CD8^+^, T cells migrate to the tumor microenvironment and release cytokines that induce killing of tumor cells. Adapted from BioRender.com. (Abbreviations: TCR, T cell receptor; MHC-I and MHC-II, major histocompatibility complex I and II; CD, cluster of differentiation; APC, antigen-presenting cell; IFN, interferon; IL, interleukin; TNF, tumor necrosis factor.

**Table 1 diseases-10-00060-t001:** Selected phase II/III/IV clinical trials of immune checkpoint inhibitor combinations in various human cancers. Data source ClinicalTrials.gov (www.clinicaltrials.gov, accessed on 25 June 2022).

Disease Condition	Drug Combination	Phase	Participants	Status	NCT Number
Non-small cell lung cancer	Durvalumab (anti-PDL1), Oleclumab (anti-CD73), Monalizumab (anti-NKG2A) Durvalumab (anti-PDL1), Domvanalimab (anti-TIGIT) Atezolizumab (anti-PDL1), Pemetrexed, Carboplatin, Cisplatin, Gemcitabine, Paclitaxel	III III III	999 860 114	Recruiting Recruiting Recruiting	NCT05221840 NCT05211895 NCT05047250
Extensive stage small cell lung cancer	Atezolizumab (anti-PDL1), Chemotherapy (Carboplatin-etoposide)	III	200	Not yet recruiting	NCT05468489
Metastatic non-small cell lung cancer	Pembrolizumab (anti-PD1), Datopotamab deruxtecan (ADC with TROP2 Ab and Dato-DXd, DS-1062a) Ipilimumab (anti-CTLA-4), Nivolumab (anti-PD1)	III III	740 265	Recruiting Active, not recruiting	NCT05215340 NCT03469960
Squamous cell non-small cell lung cancer	Sintilimab (anti-PD1), Carboplatin, Albumin-Bound Paclitaxel	III	236	Not yet recruiting	NCT05429463
Head and Neck squamous cell carcinoma	Monalizumab (anti-NKG2A), Cetuximab (anti-EGFR)	III	624	Recruiting	NCT04590963
Nasopharyngeal carcinoma	PD-1 antibody, Capecitabine	III	556	Recruiting	NCT05342792
Gastric cancer	SHR-1701 (bifunctional antibody against PD-L1 and TGF-βRII)	III	896	Enrolling by invitation	NCT05149807
Colorectal carcinoma	Sintilimab (anti-PD1), Oxaliplatin, Capecitabine	III	323	Recruiting	NCT05236972
Anal cancer	Sintilimab (anti- PD-1), Chemoradiotherapy	III	102	Recruiting	NCT05374252
Metastatic urothelial cancer	Avelumab (anti-PDL1), Cabozantinib S-malate	III	654	Recruiting	NCT05092958
Renal cell carcinoma	Nivolumab (anti-PD1), Tivozanib	III	326	Recruiting	NCT04987203
Acute myeloid leukemia	Magrolimab (anti-CD47), Venetoclax, Azacitidine	III	432	Recruiting	NCT05079230
Relapsed or refractory myeloma	Talquetamab (bispecific Ab binding CD3 and GPRC5D), Daratumumab (anti-CD38), Pomalidomide, Dexamethasone	III	810	Not yet Recruiting	NCT05455320
Recurrent myeloma	Satuximab (anti-CD38), Dexamethasone, Pomalidomide	III	534	Recruiting	NCT05405166
Melanoma	Fianlimab (anti-LAG3), Cemiplimab (anti-PD1), Pembrolizumab (anti-PD1) Nivolumab (anti-PD1) (subcutaneously versus intravenous), rHuPH20	III III	1100 286	Recruiting Recruiting	NCT05352672 NCT05297565
Breast cancer	Inetetamab (anti-HER2), Toripalimab (anti-PD1), Albumin-Bound Paclitaxel	IV	70	Not yet Recruiting	NCT05291910

**Table 2 diseases-10-00060-t002:** Selected CAR T therapies in clinical trials. Data source ClinicalTrials.gov (www.clinicaltrials.gov, accessed on 25 June 2022).

Disease Condition	Drug Combinations	Phase	Participants	Status	NCT Number
Breast cancer	4SCAR T cells (CAR-T cells targeting Her2, GD2, and CD44v6)	I/II	100	Recruiting	NCT04430595
Acute myeloid leukemia	CAR-T CD19 CD7 CAR-T cells	II/III I/II	10 108	Recruiting Recruiting	NCT04257175 NCT04599556
Multiple myeloma	CAR-T cell targeting B-cell maturation antigen (BCMA), Bortezomib, Dexamethasone, Lenalidomide, Cyclophosphamide, Fludarabine. JNJ-68284528 (cilta-cel), Pomalidomide, Bortezomib, Dexamethasone, Daratumumab JNJ-68284528 (ciltacabtagene autoleucel [cilta-cel], Bortezomib, Lenalidomide, Dexamethasone, Cyclophosphamide, Fludarabine, Daratumumab	III III III	650 419 750	Recruiting Active not yet recruiting Not yet recruiting	NCT04923893 NCT04181827 NCT05257083
B cell lymphoma	CAR-T-CD19, BTK inhibitor, Fludarabine, Cyclophosphamide	III	24	Recruiting	NCT05020392
B Cell malignancies	CD19/CD22-CAR-T cells, fludarabine, cyclophosphamide	I/II	146	Not yet Recruiting	NCT05442515
Solid tumor	CLDN6 CAR-T, CLDN6 RNA-LPX	I/II	96	Recruiting	NCT04503278
Pancreatic cancer	CD276 CAR-T cells	I/II	10	Recruiting	NCT05143151
Gastric cancer, Pancreatic cancer	CT041 (CAR-T cells targeting claudin18.2)	I/II	110	Recruiting	NCT04404595
Prostate cancer	4SCAR-PSMA T cells [CAR-T cells targeting Prostate-specific membrane antigen (PSMA)]	I/II	100	Recruiting	NCT04429451
CD44v6 positive cancers (squamous cell carcinomas, adenocarcinomas, melanoma, lymphoma)	4SCAR-CD44v6 [CAR-T cells targeting CD44v6]	I/II	100	Recruiting	NCT04427449

**Table 3 diseases-10-00060-t003:** List of some of the selected ongoing cancer vaccine clinical trials that are in phase II/III/IV stages. Data source ClinicalTrials.gov (www.clinicaltrials.gov, accessed on 25 June 2022).

Disease Condition	Drug Combination	Phase	Participants	Status	NCT Number
Breast cancer	AST-301[pNGVL3-hICD (DNA vaccine against HER2)], rhuGM-CSF, Pembrolizumab Adagloxad simolenin, OBI-821 (Vaccine with tumor-associated antigen Globo H linked to KLH).	II III	146 668	Recruiting Recruiting	NCT05163223 NCT03562637
Cervical cancer	Recombinant Human Papillomavirus Bivalent Vaccine, Recombinant Human Papillomavirus Nonavalent Vaccine, Diphtheria Toxoid/Tetanus Toxoid/Acellular Pertussis Vaccine Cecolin^®^ (bivalent HPV vaccine) Gardasil^®^ (HPV 9-valent Vaccine) Gardasil-9 (9-valent HPV Vaccination)	IV III III	5000 1025 1220	Enrolling by invitation Active, not recruiting Not yet recruiting	NCT05237947 NCT04508309 NCT03848039
Colorectal cancer	GRT-C901, GRT-R902[Chimpanzee adenovirus vector (ChAdV)twenty tumor-specific neoantigens (TSNAs)], Atezolizumab, Ipilimumab, Fluoropyrimidine, Bevacizumab, Oxaliplatin	II/III	665	Recruiting	NCT05141721
Bladder cancer	Bacillus Calmette-Guérin (BCG) Bacillus Calmette Guerin (BCG), PF-06801591(anti-PD1 mAb)	III III	32 1160	Active not yet recruiting Recruiting	NCT04806178 NCT04165317
Liver cancer	GP96 (Heat Shock Protein-Peptide Complex Vaccine)	II/III	80	Not yet Recruiting	NCT04206254
Acute myeloid leukemia	DSP-7888 [vaccine with two synthetic peptides derived from Wilms’ tumor 1 (WT1)	II	100	Recruiting	NCT04747002
Non-small cell lung cancer	UCPVax [vaccine with two peptides from hTERT], Nivolumab	II	111	Recruiting	NCT04263051
Glioblastoma multiforme	ADCTA-SSI-G1 (Autologous Dendritic Cell/Tumor Antigen)	III	118	Recruiting	NCT04277221

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
