# Peer review of "Emerging Trends in Immunotherapy for Cancer"

_diseases, 2022, doi:10.3390/diseases10030060_

Round 1

Reviewer 1 Report

An excellent review covering immunotherapeutics and their uses, expertly written and well described.

A few minor points and suggested changes.

Section 3.1. Adaptive immune checkpoints - Please include the full names for PD1 and PD-L1.  Also please include descriptions of other immune checkpoints including OX40, VISTA etc.

Section 3.4. Limitations and challenges of ICI therapy - Please include a comment on the durability of response to each therapy

Section 4.1. Limitations and challenges of CAR T therapy - Please include the full name for B cell maturation antigen

Section 7. Oncolytic viruses - Please include a short description of the mechanism of action for Delytact as per the previous examples

Author Response

Reviewer-1

We thank Reviewer-1 for reviewing this manuscript and appreciating our work. As suggested, we have now incorporated all the changes indicated by Reviewer-1 in the revised version of the manuscript.

Comments and Suggestions for Authors:

 #1 An excellent review covering immunotherapeutics and their uses, expertly written and well described.

Authors' response: Thank you for appreciating our manuscript.

A few minor points and suggested changes.

#2. Section 3.1. Adaptive immune checkpoints - Please include the full names for PD1 and PD-L1.  Also please include descriptions of other immune checkpoints including OX40, VISTA etc.

Authors' response: We have included full names for PD-1 and PDL-1, and incorporated descriptions of OX40 and VISTA (sections 3.1.7 and 3.1.8). Thanks for the suggestion. 

#3. Section 3.4. Limitations and challenges of ICI therapy - Please include a comment on the durability of response to each therapy

 Authors' response: As suggested we have now included a brief description of the durability of response to ICI therapies (section 3.4).

 Section 4.1. Limitations and challenges of CAR T therapy - Please include the full name for B cell maturation antigen

Authors' response: Included as indicated (section 4.5)

 Section 7. Oncolytic viruses - Please include a short description of the mechanism of action for Delytact as per the previous examples

Authors' response: As suggested we have now included a brief description of the mechanism of action for Delytact (Section 7).

Reviewer 2 Report

Over the past 10 years, the importance of immunology in understanding cancer has multiplied. 100 years ago, immunology and cancer were brought together by Paul Ehrlich.

Today, physicians have a plethora of therapeutic options and were overwhelmed by the present possibilities.  

This review presents all aspects of immunotherapy for cancer and thus provides a valuable overview of the existing possibilities. It is comprehensive and, in my opinion, has no gaps. The figures and tables facilitate the understanding of the dense text.

I would like to raise two minor points related to the structure:

(1) Why line 444 “CAR T cells and line 507 “CAR NK cells are not numbered?

(1) The structure with many sub-points makes the reading of this 38-page review difficult. A table of contents on the first page would be helpful (decision of the journal).

Author Response

Reviewer-2

We thank Reviewer-2 for reviewing this manuscript and appreciating our work. As suggested, we have now incorporated all the changes indicated by Reviewer-1 in the revised version of the manuscript.

Comments and Suggestions for Authors

#1. Over the past 10 years, the importance of immunology in understanding cancer has multiplied. 100 years ago, immunology and cancer were brought together by Paul Ehrlich. Today, physicians have a plethora of therapeutic options and were overwhelmed by the present possibilities. This review presents all aspects of immunotherapy for cancer and thus provides a valuable overview of the existing possibilities. It is comprehensive and, in my opinion, has no gaps. The figures and tables facilitate the understanding of the dense text.

Authors' response: Thank you for your kind appraisal.

I would like to raise two minor points related to the structure: 

  • Why line 444 “CAR T cells and line 507 “CAR NK cells are not numbered?

Authors' response: We have included the numbering in the revised version.

  • The structure with many sub-points makes the reading of this 38-page review difficult. A table of contents on the first page would be helpful (decision of the journal).

Authors' response: Thanks for the thoughtful suggestion. We will discuss with the journal if they have a provision of including a table of contents.

Reviewer 3 Report

Evaluation

Manuscript ID: diseases-1882685

Type of manuscript: Review 
Title: Emerging Trends in Immunotherapy for Cancer

The authors describe a good review on cancer immunotherapy.

Congratulations for the effort and work.

However, I have one comment for the authors.

1.- line 437: “Three types of adoptive cell therapies have been developed such as CART (chimeric antigen receptor T cells), 438 TCRs (engineered T-cell receptors), and TILs (tumor-infiltrating lymphocytes)”.

CART therapy is mentioned extensively, however, the other two types of adoptive therapy are not described. Although CAR-T is a leader in adoptive immunotherapy, the other two types should be briefly described.

Author Response

Reviewer-3

We thank Reviewer-3 for reviewing this manuscript and appreciating our work. As suggested, we have now incorporated all the changes indicated by Reviewer-3 in the revised version of the manuscript.

Comments and Suggestions for Authors 

#The authors describe a good review on cancer immunotherapy.  Congratulations for the effort and work.

Authors' response: Thank you for your kind appraisal.

However, I have one comment for the authors.

1.- line 437: “Three types of adoptive cell therapies have been developed such as CART (chimeric antigen receptor T cells), 438 TCRs (engineered T-cell receptors), and TILs (tumor-infiltrating lymphocytes)”. CART therapy is mentioned extensively, however, the other two types of adoptive therapy are not described. Although CAR-T is a leader in adoptive immunotherapy, the other two types should be briefly described.

Authors' response: We appreciate your thoughtful advice. In the revised version we have now included a short description of TILs and TCR therapy (sections 4.1 and 4.2)